# On the Alignment Between Supervised and Self-Supervised Contrastive Learning

**Achleshwar Luthra**     **Priyadarsi Mishra**     **Tomer Galanti**
Department of Computer Science and Engineering
Texas A&M University
`{luthra,priyadarsimishra,galanti}@tamu.edu`

## ABSTRACT

Self-supervised contrastive learning (CL) has achieved remarkable empirical success, often producing representations that rival supervised pre-training on downstream tasks. Recent theory explains this by showing that the CL loss closely approximates a supervised surrogate, Negatives-Only Supervised Contrastive Learning (NSCL) loss, as the number of classes grows. Yet this loss-level similarity leaves an open question: *Do CL and NSCL also remain aligned at the representation level throughout training, not just in their objectives?*

We address this by analyzing the representation alignment of CL and NSCL models trained under shared randomness (same initialization, batches, and augmentations). First, we show that their induced representations remain similar: specifically, we prove that the similarity matrices of CL and NSCL stay close under realistic conditions. Our bounds provide high-probability guarantees on alignment metrics such as centered kernel alignment (CKA) and representational similarity analysis (RSA), and they clarify how alignment improves with more classes, higher temperatures, and its dependence on batch size. In contrast, we demonstrate that parameter-space coupling is inherently unstable: divergence between CL and NSCL weights can grow exponentially with training time. Finally, we validate these predictions empirically, showing that CL–NSCL alignment strengthens with scale and temperature, and that NSCL tracks CL more closely than other supervised objectives. This positions NSCL as a principled bridge between self-supervised and supervised learning. Our code is available at dlfundamentals.github.io/cl-nscl-representation-alignment.

## 1 INTRODUCTION

Self-supervised learning (SSL) has become the dominant approach for extracting transferable representations from large-scale unlabeled data. By leveraging training signals derived directly from the data, SSL methods avoid costly annotation while producing features that generalize across modalities, from vision (Chen et al., 2020; He et al., 2020; Zbontar et al., 2021; He et al., 2022; Oquab et al., 2024) to language (Gao et al., 2021; Reimers & Gurevych, 2019), speech (Schneider et al., 2019; Baevski et al., 2020; Hsu et al., 2021; Baevski et al., 2022), and vision–language (Radford et al., 2021; Jia et al., 2021; Zhai et al., 2023; Tschannen et al., 2025). Among SSL approaches, *contrastive learning (CL)* has been particularly successful: methods such as SimCLR (Chen et al., 2020), MoCo (He et al., 2020; Chen et al., 2021b), and CPC (van den Oord et al., 2019) train encoders by pulling together augmented views of the same input while pushing apart other samples. This simple principle has yielded state-of-the-art performance, often rivaling or surpassing supervised pre-training.

Despite this empirical success, a central puzzle remains: why does CL recover features so well aligned with semantic class boundaries? CL models often support nearly supervised-level downstream performance (Amir et al., 2022; Ben-Shaul et al., 2023; Weng et al., 2025), suggesting that supervision is somehow implicit in the objective. Recent theoretical progress sheds light on this: Luthra et al. (2025) showed that the CL objective closely approximates a supervised variant, *Negatives-Only Supervised Contrastive Learning (NSCL)*, where same-class samples are excluded from the denominator. Their analysis established that the CL–NSCL *losses* converge as the number of classes grows, and further characterized the geometry of NSCL minimizers and their linear probe performance. These results indicate that CL carries a supervised-like signal at the *loss level*.

Yet this view leaves a crucial question unresolved:

> ***Do contrastive and supervised contrastive models remain***
> ***aligned throughout training, not just at the level of their objectives?***

Loss-level similarity does not guarantee that optimization paths coincide. In principle, differences in curvature, gradient noise, or learning rate schedules could amplify small loss discrepancies, causing stochastic gradient descent (SGD) trajectories to diverge. Thus, it remains unclear whether CL merely converges to a solution *similar* to NSCL, or whether their parameter and representations remain

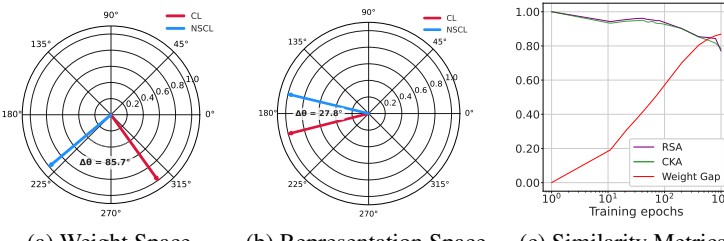

(a) Weight Space   (b) Representation Space   (c) Similarity Metrics

Figure 1: **Comparison of learning dynamics for CL and NSCL models.** (a) Weight space vectors show divergent paths ($85.7°$ apart). (b) In contrast, representation space vectors for a target class show high alignment ($27.8°$ apart). (c) This is confirmed over training epochs, where representational similarity (CKA, RSA) remains high while the weight gap increases (see figure details in App. B).

coupled across training. While some preliminary empirical results (Grigg et al., 2021) provide evidence that supervised and self-supervised models learn fairly well-aligned representations geometrically, it is not clear to what extent this alignment holds, under what conditions it arises, and what factors control the alignment between the two regimes.

**Contributions.** In this work, we theoretically and systematically study the alignment between CL and NSCL under shared randomness (same initialization, mini-batches, and augmentations):

- **From drift to metrics.** The similarity control yields explicit, high-probability *lower bounds* on linear CKA and RSA at every epoch, showing that CL and NSCL representations remain nontrivially aligned and that the certified alignment tightens as $C$ and $B$ grow and as $\tau$ increases (Cors. 1–2). For completeness, we also bound parameter drift under $\beta$-smoothness (Thm. 2), which can grow exponentially even when representations remain aligned.

- **Conceptual contribution.** Our results provide a conceptual framework for what CL optimizes during training. We (i) identify NSCL as the supervised objective whose representations and training trajectories are most tightly coupled to those of CL—without claiming that NSCL is the strongest supervised baseline in terms of top-1 accuracy—and (ii) shift the focus from guarantees on downstream classification accuracy to *geometric* alignment between supervised and self-supervised representations. Whereas prior work shows that minimizing self-supervised losses can yield good downstream classifiers under generative assumptions (e.g., (Arora et al., 2019; Tosh et al., 2021; Saunshi et al., 2022; Awasthi et al., 2022; HaoChen & Ma, 2023)), our analysis instead characterizes when CL and NSCL induce similar similarity structures, a perspective that is particularly relevant for tasks that depend on representation geometry, such as interpretability and image segmentation.

- **Empirical validation.** We validate our theory with experiments on CIFAR-10/100, Tiny-ImageNet, mini-ImageNet, and ImageNet-1K. We find that (i) CL–NSCL alignment strengthens with more classes and higher temperatures as well as correlates with the bound's dependence on the batch size; and (ii) NSCL aligns with CL more strongly than other supervised learning methods (such as cross-entropy minimization and supervised contrastive learning (SCL) (Khosla et al., 2020)).

## 2 RELATED WORK

A large body of work has sought to explain the success of contrastive learning (CL) from different perspectives. Early accounts linked CL to mutual information maximization between views of the same input (Bachman et al., 2019), though subsequent analyses showed that enforcing mutual information constraints too strongly can degrade downstream performance (McAllester & Stratos, 2020; Tschannen et al., 2020). A different line of work formalizes CL in terms of *alignment* and

*uniformity* properties of the representation space (Wang & Isola, 2020; Wang & Liu, 2021; Chen et al., 2021a), capturing how positives concentrate while negatives spread across the sphere. These geometric criteria, while intuitive, do not fully explain how samples from different semantic classes are organized under CL training.

To address this, several papers have studied the ability of CL to recover latent clusters and semantic structures (Arora et al., 2019; Tosh et al., 2021; Zimmermann et al., 2021; Ash et al., 2022; Nozawa & Sato, 2021; HaoChen et al., 2021; 2022; Shen et al., 2022; Wang et al., 2022; Awasthi et al., 2022; Bao et al., 2022). Most of these results rely on restrictive assumptions, such as conditional independence of augmentations given cluster identity (Arora et al., 2019; Tosh et al., 2021; Saunshi et al., 2022; Awasthi et al., 2022). To weaken such assumptions, HaoChen & Ma (2023) proposed analyzing spectral contrastive objectives that encourage cluster preservation without requiring augmentation connectivity, while Parulekar et al. (2023) showed that InfoNCE itself learns cluster-preserving embeddings when the hypothesis class is capacity-limited.

Another perspective comes from linking CL to supervised learning. For instance, Balestriero & LeCun (2024) showed that in linear models, self-supervised objectives such as VicReg coincide with supervised quadratic losses. In addition, Luthra et al. (2025) established an explicit coupling between the InfoNCE contrastive loss and a supervised variant that removes positives from the denominator. In contrast to prior results, these bounds are label-agnostic, architecture-independent, and hold uniformly throughout optimization. In a related vein, Lee (2025) formulate self-supervised contrastive learning as an approximation to supervised prototype-based objectives, deriving a balanced contrastive loss closely related to InfoNCE. On the representation-level alignment side, Grigg et al. (2021) provided empirical evidence that supervised and self-supervised trained models learn fairly geometrically aligned representations.

Beyond clustering and supervision, other theoretical studies have examined different aspects of CL: feature learning dynamics in linear and shallow nonlinear networks (Tian, 2022; Ji et al., 2023; Wen & Li, 2021; Tian, 2023), the role and optimality of augmentations (Tian et al., 2020; Feigin et al., 2025), the projection head (Gupta et al., 2022; Gui et al., 2023; Xue et al., 2024; Ouyang et al., 2025), sample complexity (Alon et al., 2024), and strategies to reduce batch-size requirements (Yuan et al., 2022). Finally, several works explore connections between contrastive and non-contrastive SSL paradigms (Wei et al., 2021; Balestriero & LeCun, 2022; Lee et al., 2021; Garrido et al., 2023; Shwartz-Ziv et al., 2023).

## 3 PROBLEM SETUP

We work with a dataset $S = \{(x_i, y_i)\}_{i=1}^N \subset \mathcal{X} \times [C]$ (with $C$ classes), where $[C] = \{1, \ldots, C\}$ and each class $c$ contributes $n_c$ examples. Here, $N = \sum_c n_c$ is the total number of samples and let $\pi_c = n_c/N$. An encoder $f_w : \mathcal{X} \to \mathbb{R}^d$ with parameters $w \in \mathbb{R}^p$ maps inputs to embeddings. Similarity is measured by a bounded function $\text{sim} : \mathbb{R}^d \times \mathbb{R}^d \to [-1, 1]$; throughout our experiments we use cosine similarity on $\ell_2$-normalized embeddings, $\text{sim}(u, v) = \langle u, v \rangle / (\|u\| \|v\|)$.

Data augmentations are modeled by a Markov kernel $\alpha(\cdot \mid x)$ on $\mathcal{X}$: given $x$, we draw an independent view $x' \sim \alpha(x)$. Unless stated otherwise, augmentation draws are independent across samples, across repeated views of the same sample, and across training steps. We write $x_i' \sim \alpha(x_i)$ for a single view and $(x_i^{(1)}, x_i^{(2)}) \overset{\text{i.i.d.}}{\sim} \alpha(x_i)$ for two views of the same input.

Fix a batch size $B \in \mathbb{N}$. A batch is a multiset $\mathcal{B} = \{(x_i, x_i', y_i)\}_{i=1}^B$ sampled with replacement from $S$, with independent augmentations $x_i' \sim \alpha(x_i)$. For each element in the batch, define $z_i := f_w(x_i)$ and $z_i' := f_w(x_i')$. For any anchor triple $(x_i, x_i', y_i) \in \mathcal{B}$, define the per-anchor CL loss and the CL batch loss as

$$\ell_i^{\text{CL}}(w; \mathcal{B}) := -\log \frac{\exp(\text{sim}(z_i, z_i')/\tau)}{\displaystyle\sum_{\substack{t=1 \\ t \neq i}}^B \exp(\text{sim}(z_i, z_t)/\tau) + \exp(\text{sim}(z_i, z_t')/\tau)},$$

$$\bar{\ell}_{\mathcal{B}}^{\text{CL}}(w) := \frac{1}{B} \sum_{i=1}^B \ell_i^{\text{CL}}(w; \mathcal{B}).$$

For the same realized batch $\mathcal{B}$, define the negative index set $I_i^- := \{j \in \{1, \ldots, B\} : y_j \neq y_i\}$ and the corresponding negative subset $\mathcal{B}_i^- := \{(x_j, x_j', y_j) : j \in I_i^-\}$. The NSCL per-anchor and batch losses are

$$\ell_i^{\mathrm{NSCL}}(w; \mathcal{B}_i^-) \ := \ -\log \frac{\exp\big(\mathrm{sim}(z_i, z_i')/\tau\big)}{\displaystyle\sum_{j \in I_i^-} \Big[\exp\big(\mathrm{sim}(z_i, z_j)/\tau\big) + \exp\big(\mathrm{sim}(z_i, z_j')/\tau\big)\Big]},$$

$$\bar{\ell}_{\mathcal{B}}^{\mathrm{NSCL}}(w) \ := \ \frac{1}{B} \sum_{i=1}^{B} \ell_i^{\mathrm{NSCL}}(w; \mathcal{B}_i^-).$$

Prior work (Luthra et al., 2025) shows that the CL–NSCL *loss gap* is uniformly $\mathcal{O}(1/C)$, but what we ultimately care about is whether the *embeddings* align. To quantify representation similarity we use linear Centered Kernel Alignment (CKA) and Representation Similarity Analysis (RSA) (Kornblith et al., 2019; Kriegeskorte et al., 2008) defined on cosine-similarity matrices: for $N$ common inputs with embeddings $Z = \{z_i\}_{i=1}^N$ and $Z' = \{z_i'\}_{i=1}^N$, let $\Sigma(Z)_{ij} = \cos(z_i, z_j)$ and $H = I - \frac{1}{N}\mathbf{1}\mathbf{1}^\top$; linear CKA is

$$\mathrm{CKA}(Z, Z') \ = \ \frac{\langle H\Sigma(Z)H, \ H\Sigma(Z')H \rangle_F}{\|H\Sigma(Z)H\|_F \, \|H\Sigma(Z')H\|_F},$$

and RSA is the *Pearson* correlation between the (upper–triangular) off–diagonal entries of the dissimilarity matrices $\mathrm{RDM}(Z) = \mathbf{1}\mathbf{1}^\top - \Sigma(Z)$ and $\mathrm{RDM}(Z') = \mathbf{1}\mathbf{1}^\top - \Sigma(Z')$:

$$\mathrm{RSA}(Z, Z') \ = \ \mathrm{Corr}\left(\mathrm{vec}_\triangle(\mathrm{RDM}(Z)), \mathrm{vec}_\triangle(\mathrm{RDM}(Z'))\right),$$

where $\mathrm{vec}_\triangle$ stacks the upper–triangular entries $(i < j)$ column-wise.

This raises the following question: ***Beyond a small objective gap, does training CL and NSCL actually lead to similar representations (e.g., high CKA/RSA)?***

In the spirit of Thm. 1 of Luthra et al. (2025), we prove that when two runs use shared randomness (same initialization, mini-batches, and augmentations), the per-step gradient mismatch is uniformly bounded (Lem. 7). Similarly, we show that the CL and NSCL similarity matrices remain close throughout training (Thm. 1), which yields explicit CKA/RSA lower bounds (Cors. 1-2).

## 4 THEORY

We examine how contrastive learning (CL) and negatives-only supervised contrastive learning (NSCL) co-evolve when initialized identically and trained with the same mini-batches and augmentations. While one might first attempt to study their trajectories in parameter space, such an approach quickly breaks down: without strong assumptions on the loss landscape (e.g., convexity or strong convexity), small reparameterizations can distort distances, and nonconvex dynamics cause parameter drift to grow uncontrollably over time (see App. C). For this reason, we set weight-space coupling aside and turn instead to the aspect that directly shapes downstream behavior—the *representations*—analyzing their alignment in similarity space.

### 4.1 COUPLING IN REPRESENTATION (SIMILARITY) SPACE

Let $\Sigma_t \in [-1, 1]^{N \times N}$ denote the pairwise similarity matrix of a fixed reference set at step $t$ (cosine similarity of normalized embeddings; diagonals are 1). We analyze the coupled evolution of the CL and NSCL similarities, $\Sigma_t^{\mathrm{CL}}$, and $\Sigma_t^{\mathrm{NSCL}} \in [-1, 1]^{N \times N}$ under identical mini-batches and augmentations. This representation-space view is invariant to reparameterization and directly tracks representational geometry.

**Surrogate similarity dynamics.** To make the analysis explicit, we work with a "similarity-descent" surrogate that updates only those entries touched by the current batch. For a realized mini-batch $\mathcal{B}_t = \{(x_j, x_j', y_j)\}_{j=1}^B$ (with $x_j' \sim \alpha(x_j)$), let $\bar{\ell}_{\mathcal{B}_t}^{\mathrm{CL}}(\Sigma)$ and $\bar{\ell}_{\mathcal{B}_t}^{\mathrm{NSCL}}(\Sigma)$ be the usual InfoNCE-type losses written as functions of the relevant similarity entries (with temperature $\tau > 0$). Define the batch-gradient maps

$$G_t^{\mathrm{CL}} \ := \ \nabla_\Sigma \bar{\ell}_{\mathcal{B}_t}^{\mathrm{CL}}\big(\Sigma_t^{\mathrm{CL}}\big), \qquad G_t^{\mathrm{NSCL}} \ := \ \nabla_\Sigma \bar{\ell}_{\mathcal{B}_t}^{\mathrm{NSCL}}\big(\Sigma_t^{\mathrm{NSCL}}\big),$$

setting all untouched entries to zero. The surrogate updates are

$$\Sigma_{t+1}^{\mathrm{CL}} \;=\; \Sigma_t^{\mathrm{CL}} - \eta_t\, G_t^{\mathrm{CL}}, \qquad \Sigma_{t+1}^{\mathrm{NSCL}} \;=\; \Sigma_t^{\mathrm{NSCL}} - \eta_t\, G_t^{\mathrm{NSCL}}, \tag{1}$$

with shared initialization and shared randomness (same $\mathcal{B}_t$ and augmentations).

In App. D we show that these surrogate dynamics faithfully track the similarity evolution induced by parameter-space SGD. Intuitively, for the similarity map $\Sigma(w)$ and corresponding batch loss $\bar{\ell}(w)$ (either for CL or NSCL), one SGD step $w_{t+1} = w_t - \eta_t \nabla_w \bar{\ell}(w_t)$ induces $\Sigma(w_{t+1}) - \Sigma(w_t) = -\eta_t P_t G_t + R_t$, $G_t := \nabla_\Sigma \bar{\ell}(\Sigma(w_t))$, $P_t := J_t J_t^\top$, $J_t := \partial\Sigma/\partial w|_{w_t}$, up to a second-order remainder $R_t$. Under the regularity assumptions $\|J(w)\|_{2\to 2} \le L_\Sigma$ and a quadratic Taylor bound on $\Sigma(w+\Delta w)$, together with bounded gradients and a learning-rate schedule with bounded $\sum_t \eta_t/(\tau^2 B)$ and $\sum_t \eta_t^2$, App. D shows that the induced trajectory $\widehat{\Sigma}_t := \Sigma(w_t)$ and the similarity-descent trajectory remain uniformly close. In particular, for small step sizes, sufficiently large batch $B$, and moderate temperature $\tau$, parameter-space SGD moves similarities almost as if we performed gradient descent directly in similarity space, so the surrogate dynamics faithfully track the evolution of CL and NSCL representations. We now formalize the coupling bound.

**Additional notation for high–probability factors.** Fix a training horizon $T \in \mathbb{N}$, a confidence level $\delta \in (0,1)$, and a temperature $\tau > 0$. For later use, define $\epsilon_{B,\delta} := \sqrt{\frac{1}{2B}\log\left(\frac{TB}{\delta}\right)}$ and $\Delta_{\pi,\delta}(B;\tau) := \frac{2\, e^{2/\tau}(\pi_{\max}+\epsilon_{B,\delta})}{1-\pi_{\max}-\epsilon_{B,\delta}}$ (where $\pi_{\max} = \max_c \pi_c$), and assume $\epsilon_{B,\delta} < 1 - \frac{1}{C}$ so the denominator is positive.

**Theorem 1** (Similarity-space coupling). *Fix $B, T \in \mathbb{N}$, $\delta \in (0,1)$, and temperature $\tau > 0$. Consider the coupled similarity-descent recursions equation 1 for CL and NSCL with shared initialization and shared mini-batches/augmentations. Then, with probability at least $1 - \delta$ over the draws of the mini-batches and augmentations, for any stepsizes $(\eta_t)_{t=0}^{T-1}$,*

$$\left\|\Sigma_T^{\mathrm{CL}} - \Sigma_T^{\mathrm{NSCL}}\right\|_F \;\le\; \exp\!\Big(\frac{1}{2\tau^2 B}\sum_{t=0}^{T-1}\eta_t\Big)\, \frac{1}{\tau\sqrt{B}}\Big(\sum_{t=0}^{T-1}\eta_t\Big)\Delta_{\pi,\delta}(B;\tau). \tag{2}$$

The above bound makes explicit how standard CL design choices control the discrepancy between CL and NSCL in similarity space. In particular, both the prefactor and the exponential term in equation 2 are monotone in the usual hyperparameters, so that regimes in which CL "behaves like" NSCL correspond precisely to regimes where the right-hand side of equation 2 is small. First, assuming balanced classes, a larger number of classes $C$ reduces the $1/C$ contribution inside $\Delta_{\pi,\delta}(B;\tau)$, hence decreasing the overall bound and shrinking the CL–NSCL gap. Second, increasing the batch size $B$ simultaneously reduces the concentration error $\epsilon_{B,\delta}$ and the factor $1/\sqrt{B}$, and also shrinks the coefficient $\frac{1}{2\tau^2 B}$ in the exponential, all of which act to decrease the right-hand side of equation 2 (see Fig. 5(d)). Third, increasing the temperature $\tau$ reduces the factors $\frac{1}{\tau}$ and $\frac{1}{\tau^2}$ appearing in the prefactor and exponent, again decreasing the upper bound in equation 2, consistent with the empirical trend in Fig. 4 that higher temperatures bring CL closer to NSCL. Finally, smaller learning rates $\eta_t$ (or, more generally, a smaller total step size $\sum_t \eta_t$) reduce both the prefactor $\frac{1}{\tau\sqrt{B}}\sum_t \eta_t$ and the exponent $\exp\big(\frac{1}{2\tau^2 B}\sum_t \eta_t\big)$, so more conservative optimization schedules yield a tighter coupling between CL and NSCL (see Fig. 5). Overall, Thm. 1 shows that large batches, high temperatures, and small effective step sizes—are precisely the regimes in which the similarity dynamics of CL and NSCL nearly align.

As a final note, the result in Thm. 1 is stated in terms of similarity descent, whereas in practice we use gradient descent on the network's trainable parameters. To obtain an explicit bound on the gap between the CL and NSCL similarity matrices under standard parameter-space stochastic gradient descent, we can combine Thm. 1 with twice the bound in equation 9, applying that bound once to CL and once to NSCL.

**From similarity drift to CKA/RSA guarantees.** We translate the high-probability control on the similarity drift from Thm. 1, into bounds on two standard representational metrics.

**CKA.** Recall from Sec. 3 that linear CKA (Kornblith et al., 2019) is the normalized Frobenius inner product between centered similarity matrices. $H := I - \frac{1}{N}\mathbf{1}\mathbf{1}^\top$ be the centering projector and

define centered Gram matrices $K_T^{\text{CL}} := H\Sigma_T^{\text{CL}}H$ and $K_T^{\text{NSCL}} := H\Sigma_T^{\text{NSCL}}H$. The (linear) CKA at step $T$ is $\text{CKA}_T = \frac{\langle K_T^{\text{CL}}, K_T^{\text{NSCL}}\rangle_F}{\|K_T^{\text{CL}}\|_F \|K_T^{\text{NSCL}}\|_F} \in [0,1]$. Because $\|HXH\|_F \le \|X\|_F$, any bound on $\|\Sigma_T^{\text{CL}} - \Sigma_T^{\text{NSCL}}\|_F$ controls $\|K_T^{\text{CL}} - K_T^{\text{NSCL}}\|_F$. For convenience, introduce the relative deviation $\rho_T := \frac{\|K_T^{\text{CL}} - K_T^{\text{NSCL}}\|_F}{\|K_T^{\text{CL}}\|_F}$.

**Corollary 1** (CKA lower bound). *In the setting of Thm. 1. Assume $\|K_T^{\text{CL}}\|_F > 0$. With probability at least $1 - \delta$,*

$$\text{CKA}_T \ge \frac{1 - \rho_T}{1 + \rho_T}, \qquad \rho_T \le \frac{\exp\left(\frac{1}{2\tau^2 B}\sum_{t=0}^{T-1}\eta_t\right)\frac{1}{\tau\sqrt{B}}\left(\sum_{t=0}^{T-1}\eta_t\right)\Delta_{\pi,\delta}(B;\tau)}{\|K_T^{\text{CL}}\|_F}.$$

**RSA.** Recall from Sec. 3 that RSA (Kriegeskorte et al., 2008) is the Pearson correlation between the off-diagonal entries of representational dissimilarity matrices (RDMs). Let $M = \binom{N}{2}$ and define off-diagonal RDM vectors $a_T, b_T \in \mathbb{R}^M$ by $a_T(u,v) = 1 - \Sigma_T^{\text{CL}}(u,v)$ and $b_T(u,v) = 1 - \Sigma_T^{\text{NSCL}}(u,v)$ for $u < v$. Write $\sigma_{D,T} > 0$ for the empirical standard deviation of the entries of $a_T$. The RSA score is the Pearson correlation $\text{RSA}_T = \text{Corr}(a_T, b_T)$. Zeroing the diagonal does not increase Frobenius norms, so $\|b_T - a_T\|_2 \le \|\Sigma_T^{\text{NSCL}} - \Sigma_T^{\text{CL}}\|_F$. It will be useful to measure the relative discrepancy $r_T := \frac{\|b_T - a_T\|_2}{\sqrt{M}\,\sigma_{D,T}}$.

**Corollary 2** (RSA lower bound). *In the setting of Thm. 1. Assume $\sigma_{D,T} > 0$. With probability at least $1 - \delta$,*

$$\text{RSA}_T \ge \frac{1 - r_T}{1 + r_T}, \qquad r_T \le \frac{\exp\left(\frac{1}{2\tau^2 B}\sum_{t=0}^{T-1}\eta_t\right)\frac{1}{\tau\sqrt{B}}\left(\sum_{t=0}^{T-1}\eta_t\right)\Delta_{\pi,\delta}(B;\tau)}{\sqrt{M}\,\sigma_{D,T}}.$$

These results complement the parameter–space analysis. While parameter trajectories may diverge exponentially (in the non-convex setting), the induced similarities—and hence representational metrics such as CKA and RSA—remain tightly controlled by class count, batch size, learning rate, and temperature $\tau$. The key quantity is the similarity–matrix drift $\|\Sigma_T^{\text{CL}} - \Sigma_T^{\text{NSCL}}\|_F$, which Thm. 1 bounds in two stabilizing ways.

First, the exponential factor is moderated by the $\frac{1}{\tau^2 B}$ term in the exponent. Unlike parameter space, where the growth rate scales with $\beta$, the "instability rate" in similarity space is only $\frac{1}{2\tau^2 B}$ and is therefore negligible for typical batch sizes (e.g., $B \approx 10^2$–$10^3$).

Second, the prefactor $\frac{1}{\tau\sqrt{B}}\left(\sum_t \eta_t\right)\Delta_{\pi,\delta}(B;\tau)$ decreases rapidly with batch size and class count (note $\Delta_{\pi,\delta}(B;\tau)$ shrinks with smaller $\pi_{\max}$ and grows with smaller $\tau$ through $e^{2/\tau}$). In practical regimes ($C \sim 10^3$, $B \sim 10^2$–$10^3$), this prefactor is small, making the total Frobenius gap negligible relative to the scale of the similarity matrices.

Together, these effects yield high–probability guarantees $\text{CKA}_T \ge (1 - \rho_T)/(1 + \rho_T)$ and $\text{RSA}_T \ge (1 - r_T)/(1 + r_T)$ with $\rho_T, r_T \ll 1$ in realistic conditions. Thus, even if parameters drift, the induced representations evolve in a coupled and stable manner—consistent with empirical findings that CL and NSCL remain closely aligned in practice.

**Proof idea.** We begin with a high–probability batch–composition guarantee (Cor. 3): with probability at least $1 - \delta$, every anchor's denominator contains the expected proportion of negatives up to an $\epsilon_{B,\delta}$ fluctuation. This rules out positive–heavy batches that would otherwise cause the NSCL renormalization to deviate substantially from CL. Conditioning on this event, the CL–NSCL batch–gradient gap decomposes into (i) a *reweighting error*, bounded in total variation by $\Delta_{\pi,\delta}(B;\tau)$ (Lem. 6), and (ii) a *stability term* from the dependence on the current similarities, controlled by the $\frac{1}{2\tau^2 B}$–Lipschitzness of the batch–gradient map in Frobenius norm (Lem. 2 at temperature $\tau$). Using block–orthogonality across anchors (Lem. 1), the reweighting contributions combine in quadrature, giving the per–step estimate (Lem. 8),

$$\left\|G_t^{\text{CL}}(\Sigma_t^{\text{CL}}) - G_t^{\text{NSCL}}(\Sigma_t^{\text{NSCL}})\right\|_F \le \frac{1}{\tau}\cdot\frac{\Delta_{\pi,\delta}(B;\tau)}{\sqrt{B}} + \frac{1}{2\tau^2 B}\left\|\Sigma_t^{\text{CL}} - \Sigma_t^{\text{NSCL}}\right\|_F.$$

Consequently, the similarity drift satisfies the recurrence

$$\left\|\Sigma_{t+1}^{\text{CL}} - \Sigma_{t+1}^{\text{NSCL}}\right\|_F \le \left(1 + \frac{\eta_t}{2\tau^2 B}\right)\left\|\Sigma_t^{\text{CL}} - \Sigma_t^{\text{NSCL}}\right\|_F + \eta_t\frac{1}{\tau}\cdot\frac{\Delta_{\pi,\delta}(B;\tau)}{\sqrt{B}},$$

|      | CIFAR-10 | | CIFAR-100 | | Mini-ImageNet | | Tiny-ImageNet | |
|------|------|------|------|------|------|------|------|------|
|      | NCCC | LP | NCCC | LP | NCCC | LP | NCCC | LP |
| CL   | 88.37 | 90.16 | 54.62 | 65.65 | 60.78 | 65.30 | 40.59 | 44.61 |
| NSCL | 94.47 | 94.09 | 60.14 | 68.38 | 63.92 | 72.60 | 40.76 | 45.79 |
| SCL  | 94.93 | 94.67 | 64.06 | 69.52 | 74.78 | 76.00 | 48.63 | 48.73 |
| CE   | 92.97 | 93.39 | 67.35 | 68.04 | 75.20 | 74.00 | 48.28 | 52.57 |

Table 1: Nearest Class-Center Classifier (NCCC) and Linear Probe (LP) test accuracies (%). We report the accuracies against the all-way classification task in each dataset. The models (also used in Fig. 2) were pre-trained on their respective datasets.

where the first term propagates existing error and the second injects the new discrepancy introduced at step $t$. Unrolling this recurrence (discrete Grönwall) yields

$$\|\Sigma_T^{\text{CL}} - \Sigma_T^{\text{NSCL}}\|_F \ \leq \ \exp\Big(\frac{1}{2\tau^2 B}\sum_{t=0}^{T-1}\eta_t\Big)\frac{1}{\tau\sqrt{B}}\Big(\sum_{t=0}^{T-1}\eta_t\Big)\Delta_{\pi,\delta}(B;\tau).$$

Finally, centering contracts Frobenius norms, so this control transfers directly to the centered Gram matrices, and applying standard $(1-\rho)/(1+\rho)$ and $(1-r)/(1+r)$ comparisons yields the claimed CKA/RSA lower bounds.

## 5 EXPERIMENTS

**Datasets and augmentations.** We experiment with the following standard vision classification datasets - CIFAR10 and CIFAR100 (Krizhevsky, 2009), Mini-ImageNet (Vinyals et al., 2016), Tiny-ImageNet (Han, 2020), and ImageNet-1K (Deng et al., 2009). (See App. B for details.)

**Methods, architectures, and optimizers.** For all our experiments, we have followed the Sim-CLR (Chen et al., 2020) algorithm. We use a ResNet-50 (He et al., 2016) encoder with a width-multiplier factor of 1. The projection head follows a standard two-layer MLP architecture composed of: `Linear(2048 → 2048)` → `ReLU` → `Linear(2048 → 128)`. For cross-entropy training, we attach an additional classification head `Linear(128 → C)` where $C$ is the number of classes.

For contrastive learning, we use the DCL loss that avoids positive-negative coupling during training (Yeh et al., 2022). For supervised learning, we use the following variants: Supervised Contrastive Loss (Khosla et al., 2020), Negatives-Only Supervised Contrastive Loss (Luthra et al., 2025), and Cross-Entropy Loss (Shannon, 1948). To minimize the loss, we adopt the LARS optimizer (You et al., 2017) which has been shown in (Chen et al., 2020) to be effective for training with large batch sizes. For LARS, we set the momentum to 0.9 and the weight decay to $1\mathrm{e}^{-6}$. All experiments are carried out with a batch size of $B = 1024$. The base learning rate is scaled with batch size as $0.3 \cdot \lfloor B/256 \rfloor$, following standard practice (Chen et al., 2020). We employ a warm-up phase (Goyal et al., 2017) for the first 10 epochs, followed by a cosine learning rate schedule without restarts (Loshchilov & Hutter, 2016) for the remaining epochs. All models were trained on a single node with one 94 GB NVIDIA H100 GPU.

**Evaluation metrics.** To quantitatively measure the alignment between the learned representation spaces of different models, we monitor linear CKA and RSA (check Sec. 3 for details) during training. Both CKA and RSA range from 0 to 1, where 1 indicates identical similarity structures. To manage the significant memory requirements of $N \times N$ matrices (Gram matrices for CKA, RDMs for RSA), we use a memory-efficient, chunk-wise computation strategy.

### 5.1 EXPERIMENTAL RESULTS

**Alignment analysis as a function of epochs.** To understand how representational similarity evolves, we trained a model with a CL objective and monitored its alignment (via CKA/RSA) against supervised models trained with NSCL, CE, and SCL. We find that NSCL consistently achieves the

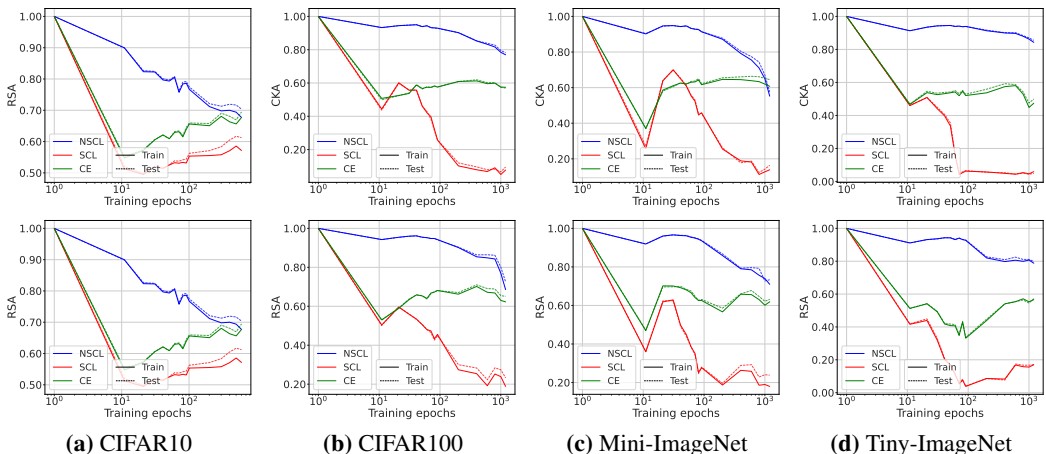

**(a)** CIFAR10     **(b)** CIFAR100     **(c)** Mini-ImageNet     **(d)** Tiny-ImageNet

Figure 2: **Alignment during training.** We train ResNet-50 models with decoupled CL, SCL, NSCL, and CE. For the first 1,000 epochs, the CL-trained model is substantially more aligned with the NSCL-trained model than with the others. However, alignment declines when training continues much longer.

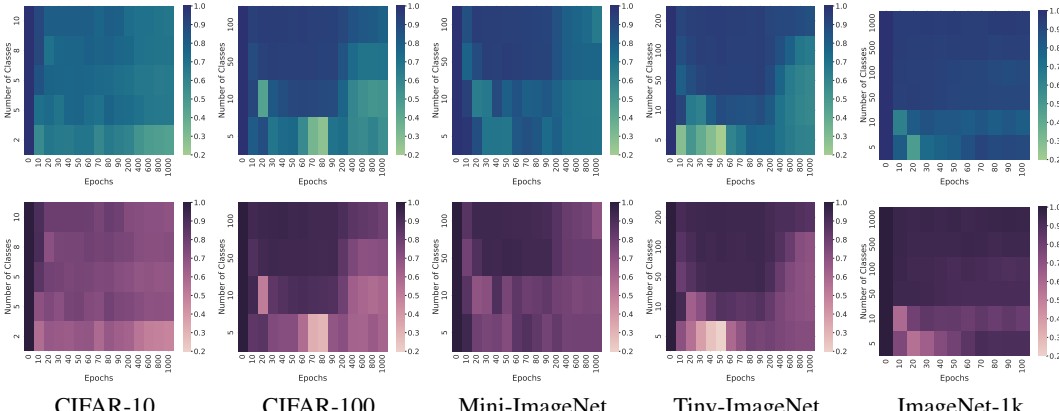

CIFAR-10     CIFAR-100     Mini-ImageNet     Tiny-ImageNet     ImageNet-1k

Figure 3: **CL–NSCL alignment (linear CKA) increases with the number of training classes.** The heatmaps show the linear CKA between CL and NSCL models. We visualize alignment on the training (top row, green) and test (bottom row, purple) sets. The y-axis indicates the number of classes ($N$) used for training, and the x-axis represents the training epoch. While alignment is consistently higher for larger $N$, it also tends to decrease as training progresses for any fixed $N$.

highest alignment with CL throughout training across multiple datasets compared to CE and SCL (see Fig. 2). For example, after 1k epochs on Tiny-ImageNet, the CL-NSCL alignment reaches a CKA of $0.87$, in contrast to just $0.043$ for CL-SCL.

Intuitively, these alignment patterns follow from how each loss shapes representation geometry. All three methods incentivize neural collapse (Papyan et al., 2020; Han et al., 2022; Zhou et al., 2022; Lu & Steinberger, 2022; Dang et al., 2024; Graf et al., 2021; Awasthi et al., 2022; Gill et al., 2023; Kini et al., 2024; Luthra et al., 2025), but differ in how directly and how quickly they drive it. NSCL is structurally closest to CL: both attract a single positive toward an anchor and repel negatives, primarily enforcing instance-level discrimination and thus inducing similar geometry. SCL, by contrast, imposes a stronger class-level constraint, explicitly pulling together augmentations of same-class samples and pushing apart different-class samples, which rapidly reduces intra-class variance and forms tight class clusters that depart from CL's instance-level structure. Cross-entropy (CE) lies between these extremes, promoting collapse more indirectly via error minimization with regularization. In the self-supervised setting, CL representations need not collapse as tightly as supervised ones, since they are learned without labels. As training enters the 10–100-epoch range,

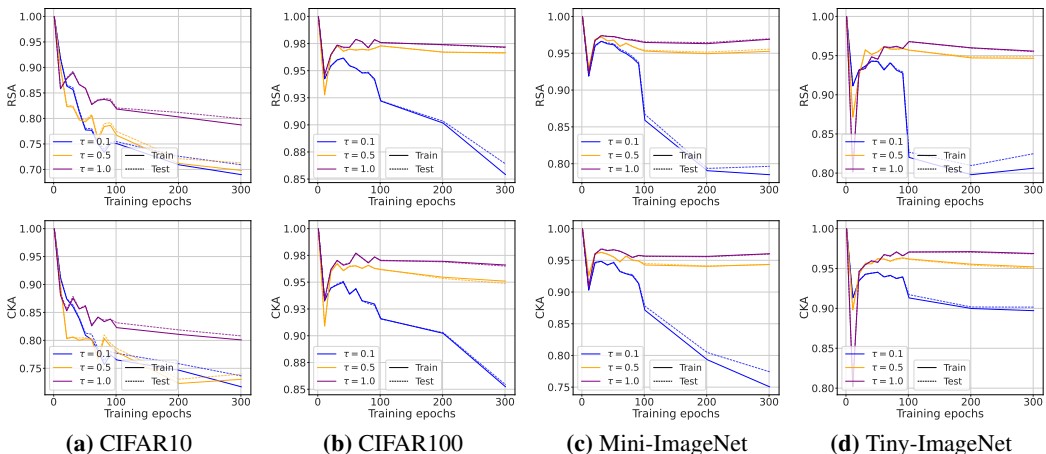

Figure 4: **Higher $\tau$ increases the CL-NSCL alignment.** The plots show RSA (top row) and CKA (bottom row) over 300 epochs. We trained CL and NSCL models with varying temperatures ($\tau \in \{0.1, 0.5, 1.0\}$) on four datasets. Across all datasets, a higher temperature $\tau = 1.0$ (shown in purple) evidently results in the highest alignment.

SCL and CE move closer to the neural collapse regime, while NSCL continues to mimic the CL label-free optimization for a longer duration, producing the evolving alignment dynamics in Fig. 2.

For completeness, along with CKA and RSA, we also report downstream performance via Nearest Class Center Classifier (Galanti et al., 2022) and Linear Probe accuracies in Tab. 1.

**Validating Thm. 1 as a function of class count.** Thm. 1 predicts that using more classes yields stronger CL–NSCL alignment. We test this via $C'$-way training: for each $C' \in [2, C]$, we train CL and NSCL on random $C'$-class subsets for 1,000 epochs (except 100 epochs for IM-1K). As shown in Fig. 3, representation similarity (RSA/CKA) increases with $C'$ across all datasets.

**Effect of temperature on alignment.** As per Thm. 1 and Cors. 1-2, CL-NSCL alignment improves with higher values of temperature ($\tau$). We empirically verify this claim by training CL and NSCL models for 300 epochs, over three different values of $\tau \in \{0.1, 0.5, 1.0\}$. Both models–CL and NSCL–are trained with same $\tau$ in each run. As shown in Fig. 4, models trained with $\tau = 1.0$ achieve higher alignment compared to models trained with lower temperatures.

**Effect of batch size on alignment.** Thm. 1 links alignment to a bound that may rise or fall with $B$ depending on how the learning rate scales. To investigate this, we vary $\eta$ with $B$ across four cases: $\eta = \frac{0.3B}{256}, \eta = \frac{0.3\sqrt{B}}{256}, \eta = \frac{0.3\sqrt[4]{B}}{256}$, and $\eta = 0.3$. Under $\mathcal{O}(B)$ scaling, CL–NSCL alignment decreases as $B$ grows, matching the theorem's implication for that scaling; for the other three cases, alignment increases with $B$, again consistent with the bound under those dependencies (see Fig. 5).

**Weight-space coupling.** We next study whether the observed alignment between representations of contrastive and supervised models is also reflected directly in their parameters. For this, we measure the average weight difference between a contrastive model and two supervised counterparts as follows: $\sum_l \frac{\|w_{\text{CL}}^l - W_{\text{sup}}^l\|_F}{0.5\,(\|w_{\text{CL}}^l\|_F + \|w_{\text{sup}}^l\|_F)}$ where $w_{\text{CL}}^l$ and $w_{\text{sup}}^l$ are weights corresponding to $l^{\text{th}}$ layer of self-supervised and supervised models respectively, and $\|\cdot\|_F$ denotes Frobenius norm. As we show in Fig. 6, for each dataset, we observe a significant divergence in weight space: both supervised models (NSCL and SCL) increasingly separate from the contrastive model as training progresses.

## 6 Conclusions, Limitations and Future Work

**Conclusions.** We studied the dynamic alignment between contrastive learning (CL) and its supervised counterpart (NSCL). By analyzing coupled SGD under shared randomness, we showed that while parameter-space trajectories may diverge exponentially, representation-space dynamics are far more stable: the similarity matrices induced by CL and NSCL remain close throughout training. This

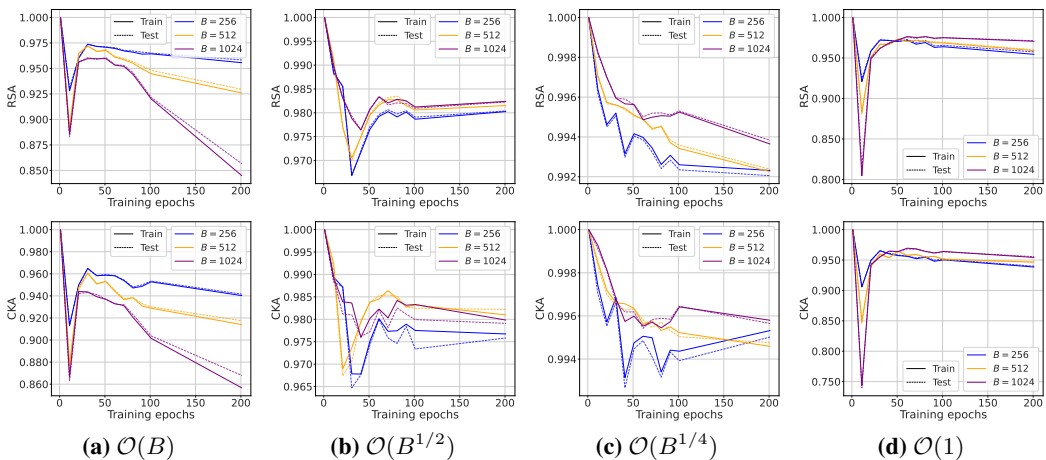

Figure 5: **Effect of batch size with scaled learning rates.** We trained CL, and NSCL models for 300 epochs with varying batch-sizes ($B \in \{256, 512, 1024\}$). For each experiment, the learning rate $\eta$ is scaled as a function of batch-size, as mentioned under each panel. For instance, the results shown in panel (b) use a learning rate of $\eta = \frac{0.3\sqrt{B}}{256}$.

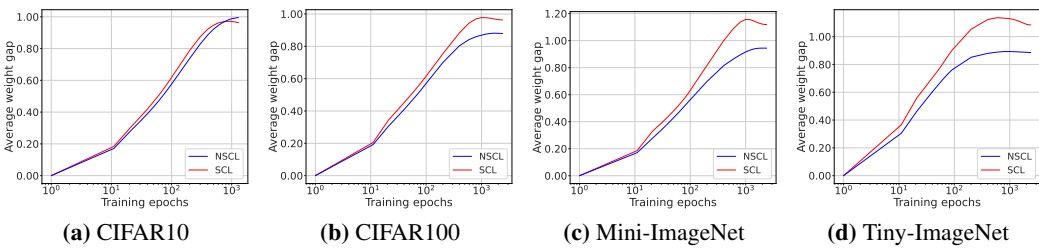

Figure 6: **Weight-space alignment quickly deteriorates.** Using the same ResNet-50 instances as in Fig. 2, we plot the average weight gap between CL and the supervised models (NSCL and SCL) across training epochs. Both supervised variants diverge from the CL model, with SCL showing a wider separation.

yields high-probability lower bounds on alignment metrics such as CKA and RSA, directly certifying representational coupling. Empirically, our experiments confirmed these trends across datasets and architectures. Together, our results highlight that the implicit supervised signal in CL is not confined to its loss function but extends throughout the entire optimization trajectory.

**Limitations.** Our theoretical bounds are structurally informative but not expected to be tight in large-scale or long-horizon regimes. As is common in machine learning theory, the guarantees are conservative worst-case bounds derived from uniform high-probability arguments, favoring generality over numerical sharpness. Many influential results in optimization and stability theory for deep learning similarly rely on loose worst-case analyses—e.g., (Bousquet & Elisseeff, 2002; Hardt et al., 2016; Mou et al., 2018; Kuzborskij & Lampert, 2017)—yet still provide useful conceptual guidance. In our setting, without additional structural assumptions (such as stronger curvature or smoothness conditions), one cannot generally expect qualitatively sharper dependence than the scaled exponential factors appearing in Thm. 1 and equation 9. Thus, while in practice the bounds are quite loose, they achieve their intended goal of identifying which parameters govern the CL–NSCL similarity gap and explaining how this gap scales with them.

**Future directions.** We view our results as a first step toward a more refined theory of self-supervised alignment. Future work could (i) derive tighter constants by exploiting data-dependent structure rather than worst-case bounds, and (ii) extend the framework to other SSL paradigms (e.g., non-contrastive methods). Improving these guarantees while retaining their stability properties would provide an even stronger theoretical bridge between supervised and self-supervised learning.

## 7 REPRODUCIBILITY STATEMENT

We have taken several steps to ensure the reproducibility of our results. All datasets used in this work (CIFAR-10/100, Tiny-ImageNet, and Mini-ImageNet) are publicly available, and we describe the data processing and augmentation pipelines in Section 3 and App. B. The theoretical results are supported by detailed proofs in App. C, D, E, where all assumptions are explicitly stated. Experimental details, including architectures, optimizers, hyperparameters, and training schedules, are reported in Section 3, with additional clarifications in the appendix. To facilitate further verification, we provide an anonymous code repository in the supplementary material that contains implementations of the CL, NSCL, and baseline objectives, along with scripts to reproduce all figures and tables in the paper. Together, these resources are intended to make both the theoretical and empirical findings fully reproducible.

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

## A  LLM Usage Statement

Large Language Models (LLMs) were used solely as an assistive tool for improving the clarity and presentation of the manuscript (e.g., editing grammar, refining phrasing). All technical content, including theoretical derivations, proofs, experimental design, and analysis, was developed entirely by the authors. No parts of the paper were written or ideated by an LLM in a way that would constitute substantive scientific contribution, and no LLM was used to generate or fabricate results.

## B  Additional Experiments

**Datasets and augmentations.** CIFAR10 and CIFAR100 both consist of 50000 training images and 10000 validation images with 10 classes and 100 classes, respectively, uniformly distributed across the dataset, i.e., CIFAR10 has 5000 samples per class and CIFAR100 has 500 samples per class. Mini-ImageNet also has 5000 test images on top of 50000 train and 10000 validation images, with 100 of 1000 classes from ImageNet-1k (Deng et al., 2009) (at the original resolution). Tiny-ImageNet contains 100000 images downsampled to $64 \times 64$, with total 200 classes from IM-1K. Each class has 500 training, 50 validation, and 50 test images.

We use standard augmentations as proposed in SimCLR (Chen et al., 2020). For experiments on Mini-ImageNet, we use the following pipeline: random resized cropping to $224 \times 224$, random horizontal flipping, color jittering (brightness, contrast, saturation: 0.8; hue: 0.2), random grayscale conversion ($p = 0.2$), and Gaussian blur (applied with probability 0.1 using a $3 \times 3$ kernel and $\sigma = 1.5$). For Tiny-ImageNet, we drop saturation to 0.4 and hue to 0.1 due to low resolution images. For CIFAR datasets, we adopt a similar pipeline with appropriately scaled parameters. The crop size is adjusted to $32 \times 32$, and the color jitter parameters are scaled to saturation 0.4, and hue 0.1.

### B.1  Experiments with the ViT Architecture

To further support the claims made in the main text, we reproduce the experiment from Fig. 2 using the ViT-Base architecture (Dosovitskiy et al., 2021). Throughout these experiments, we use the same training hyperparameters and augmentations for each dataset as in the ResNet-50 experiments. As shown in Fig. 7, the alignment between CL and supervised models exhibits the same qualitative trends observed for the ResNet-50 architecture in Fig. 2, demonstrating that the relationship between training dynamics and representational alignment is consistent across both convolutional and transformer-based models.

In addition, we repeat the experiments in Figs. 4 and 5 for the ViT-Base architecture. The corresponding results, shown in Figs. 8 and 9, closely match those obtained with ResNet-50, further reinforcing the robustness of our findings across architectures.

### B.2  Effect of number of classes on alignment

In addition to the linear CKA results reported in the main text (Fig. 3), we also evaluate representational similarity using RSA. The corresponding RSA values are presented in Fig. 10, providing a complementary perspective on alignment across varying numbers of classes. In addition, we also reproduced the results with RSA for the ViT models (Fig. 11).

### B.3  Performance-Alignment Tradeoff

The bound in Thm. 1 predicts that alignment increases with larger $\tau$. Moreover, when $\eta_t = \mathcal{O}(B)$, it suggests that alignment should decrease as $B$ grows, whereas under $\eta_t = \mathcal{O}(B^{1/4})$ it instead predicts higher alignment for larger $B$. In this experiment, we examine whether higher alignment in fact corresponds to more similar downstream accuracies. Specifically, in Figs. 12–13 we vary the parameters $\tau$ and $B$ (respectively) and plot the gap between the accuracies of the CL and NSCL models against their RSA alignment values. To obtain the accuracy measures, we perform full-shot linear probing on both the CL- and NSCL-trained models and report their test accuracies. As can be seen from the results, we consistently observe that higher alignment corresponds to a smaller gap between the accuracy rates of the CL- and NSCL-trained models. This suggests that the alignment

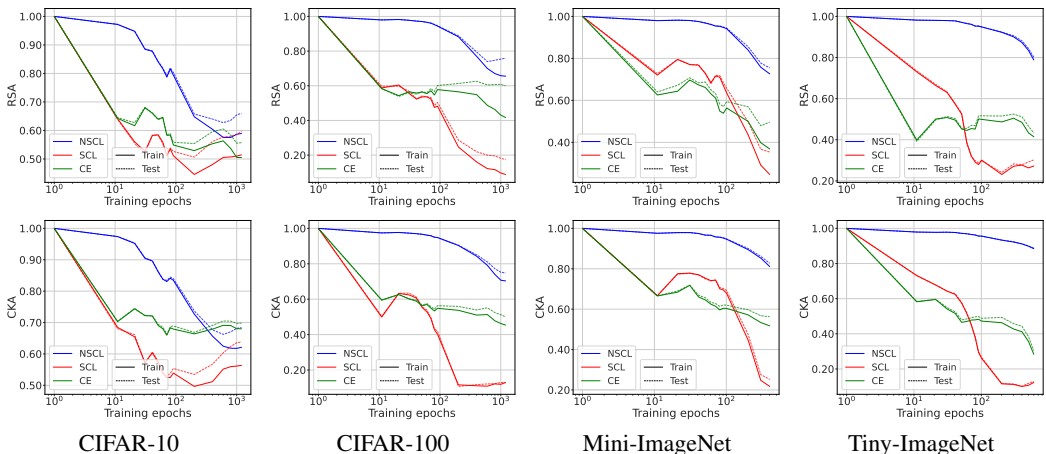

Figure 7: **Alignment during training for ViT.** We train ViT-base model with CL, NSCL, SCL and CE objectives. The alignment between CL and supervised models follow similar trends as shown for ResNet-50 in Fig. 2.

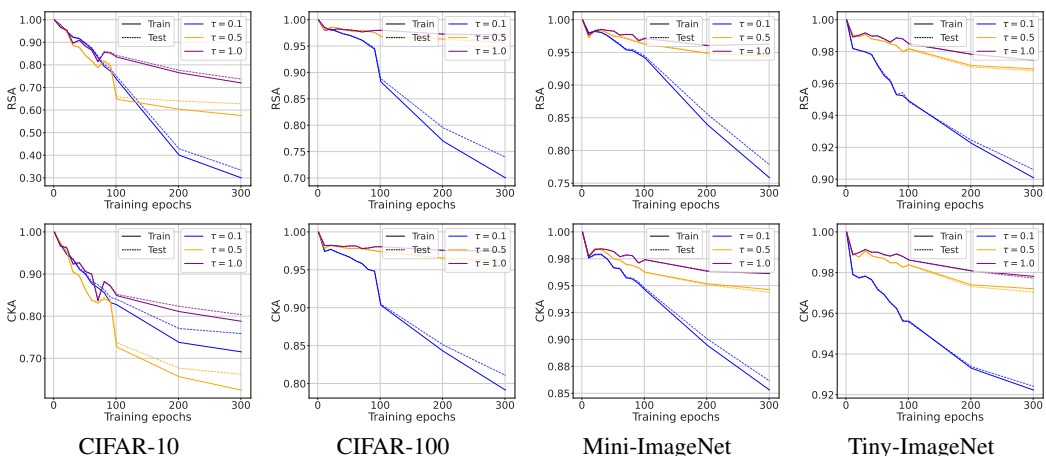

Figure 8: **Effect of the temperature ($\tau$) on CL–NSCL alignment.** We train ViT-Base models with decoupled CL and NSCL objectives using different temperature values $\tau$. All models are trained for 300 epochs. Across all datasets, alignment consistently increases as $\tau$ becomes larger.

between CL and NSCL models translates into concrete predictions about how close the models are in their performance. For completeness, we summarize these accuracy values in Tab. 2.

| | CIFAR-100 | | | Mini-ImageNet | | | Tiny-ImageNet | | |
|---|---|---|---|---|---|---|---|---|---|
| | $\tau = 0.1$ | $\tau = 0.5$ | $\tau = 1.0$ | $\tau = 0.1$ | $\tau = 0.5$ | $\tau = 1.0$ | $\tau = 0.1$ | $\tau = 0.5$ | $\tau = 1.0$ |
| CL | 65.18 | 61.62 | 58.60 | 70.30 | 70.55 | 68.21 | 44.50 | 40.41 | 35.40 |
| NSCL | 68.25 | 62.44 | 59.02 | 73.93 | 71.88 | 67.76 | 46.51 | 39.95 | 35.29 |

Table 2: Linear Probe (LP) test accuracies (%) for varying $\tau$. We train CL and NSCL ResNet-50 models for 300 epochs, and observe that the accuracy gap decreases with higher alignment between CL and NSCL models (also shown in Fig. 12).

## B.4 EXPERIMENTS WITH CLASS-IMBALANCED DATA

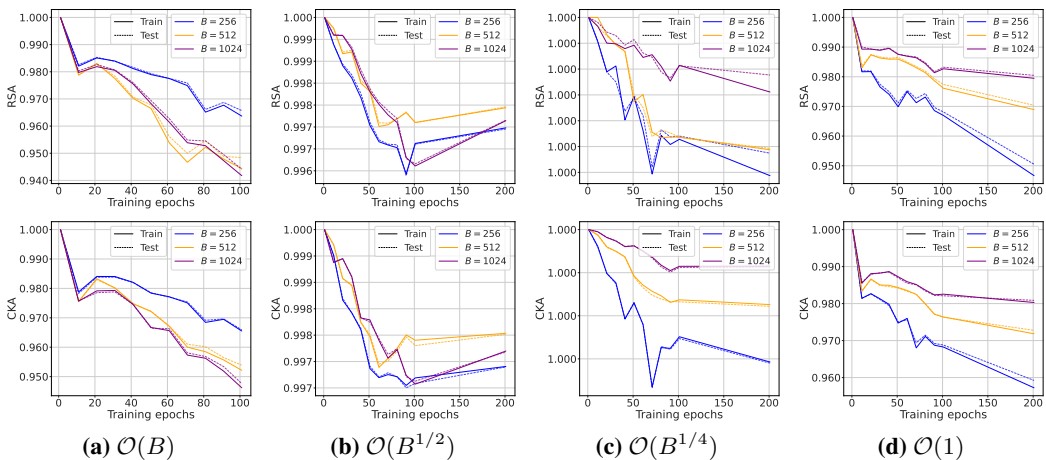

**(a)** $\mathcal{O}(B)$      **(b)** $\mathcal{O}(B^{1/2})$      **(c)** $\mathcal{O}(B^{1/4})$      **(d)** $\mathcal{O}(1)$

Figure 9: **Effect of batch size ($B$) on CL-NSCL alignment.** We follow the same learning-rate scaling strategy as for ResNet-50. The alignment trends observed when varying the batch size are similar to those for ResNet-50.

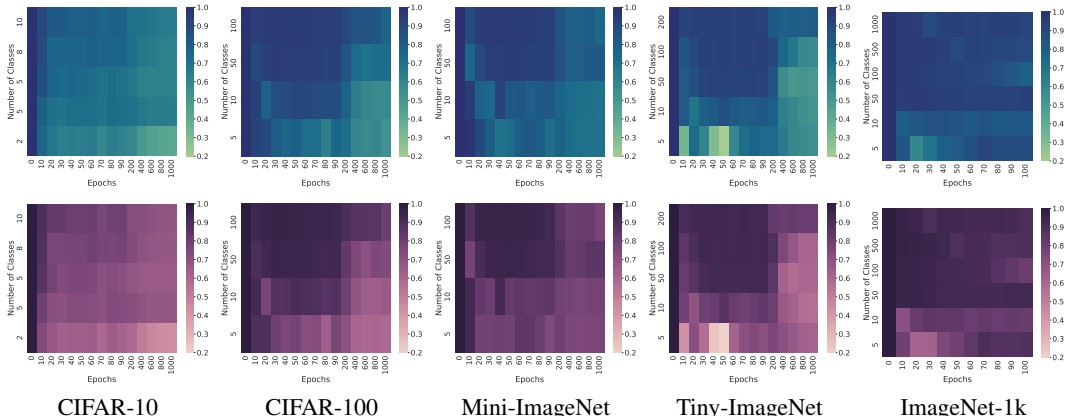

     CIFAR-10      CIFAR-100      Mini-ImageNet      Tiny-ImageNet      ImageNet-1k

Figure 10: **CL–NSCL alignment (RSA) increases with the number of training classes.** See Sec. 5.1 and Fig. 3 for experimental details.

Since our theory is tighter for relatively balanced classes, but does not require perfectly balanced data, we also evaluate it on the SVHN dataset (Netzer et al., 2011), which is well known for its pronounced class imbalance. In Fig. 14, we plot the RSA and CKA metrics between coupled CL and NSCL models trained for 300 epochs. The training hyperparameters and the data augmentations are the same as in our CIFAR-100 experiments to facilitate a direct comparison.

Despite the class imbalance in SVHN, we observe that the alignment between the two models is consistently high—indeed, it is even stronger than what we typically obtain after 1,000 epochs on CIFAR-100, which has the same number of classes. This finding suggests that substantial class imbalance does not hinder strong representational alignment from emerging between coupled CL and NSCL models, and further supports the robustness of our theoretical predictions beyond the approximately balanced setting.

### B.5 ATTENTION MAPS ALIGNMENT

**Methodology.** To analyze the self-attention maps from the frozen Vision Transformer encoder, we look into the Multi-Head Self-Attention (MHSA) mechanism of the final transformer layer ($L = 12$ for ViT-Base).

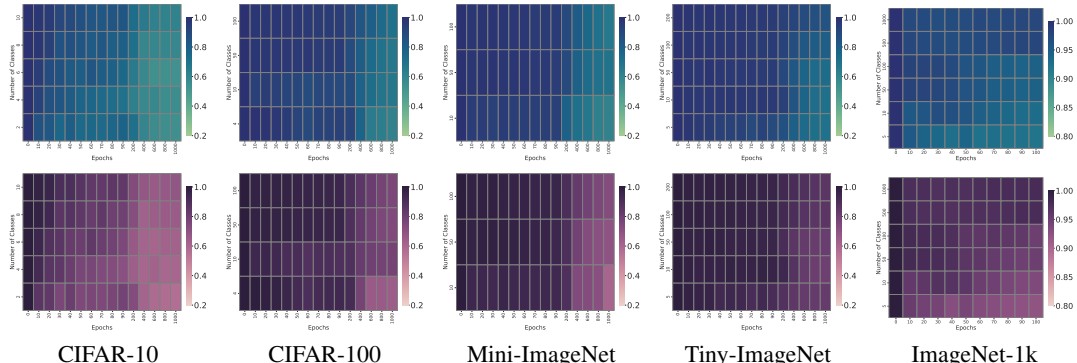

CIFAR-10    CIFAR-100    Mini-ImageNet    Tiny-ImageNet    ImageNet-1k

Figure 11: **CL–NSCL alignment (RSA) increases with the number of training classes for ViT-Base models.** The alignment increases with number of classes, and is consistent with trends observed for ResNet-50 models.

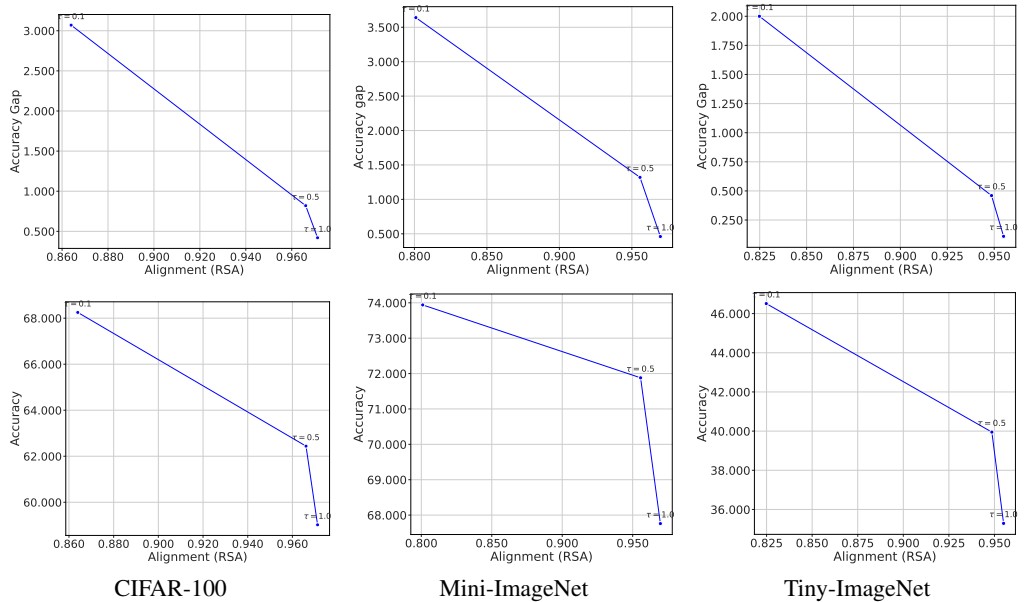

CIFAR-100    Mini-ImageNet    Tiny-ImageNet

Figure 12: **Performance vs. alignment over varying temperatures. (Top)** The gap between the linear probe accuracies of CL and NSCL ResNet-50 models (trained for 300 epochs) decreases as their alignment increases with higher temperature $(\tau)$ values. **(Bottom)** Although the accuracy gap between CL and NSCL models is correlated with the alignment of their representations, higher alignment does not necessarily imply better downstream performance, as performance remains sensitive to the choice of hyperparameters.

Let $A \in R^{H \times N \times N}$ denote the attention weights, where $H$ is the number of heads and $N$ is the number of tokens. We first average weights across all attention heads. We then extract the row corresponding to **[CLS]** token, specifically focusing on its attention to $N - 1$ image patch tokens. This vector is reshaped into a 2D grid ($14 \times 14$ for ViT-Base) to match the spatial arrangement of image patches. Finally, we upscale the low-resolution grid to original image resolution, normalize it to the range $[0, 1]$, and overlay on the input image.

**Analysis.** To quantify the structural similarity between representations of ViT models trained with decoupled CL and supervised objectives, we calculate the cosine similarity between their attention maps. As shown in Fig. 15, we track this metric across training epochs and show that NSCL consistently maintains the highest alignment with DCL compared to NSCL and CE. To strengthen our argument, we further visually illustrate this alignment in Fig. 16. The qualitative analysis align

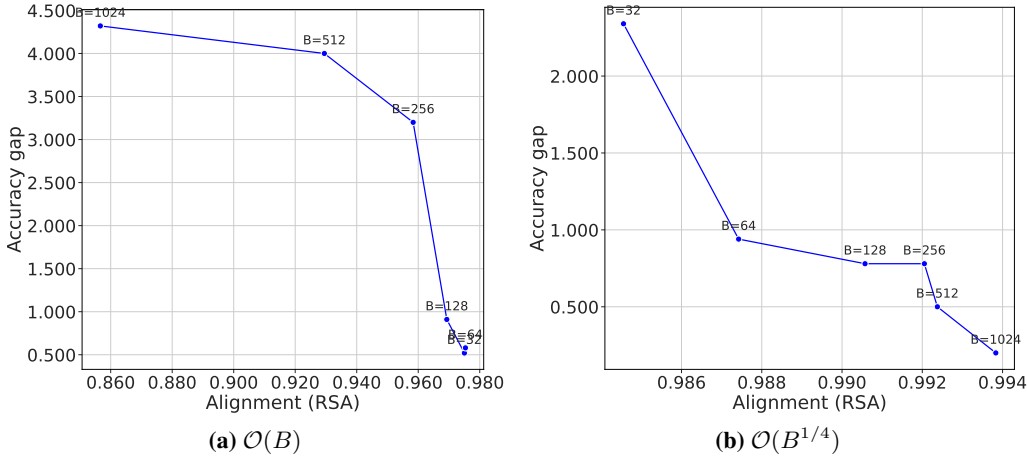

Figure 13: **Performance vs. alignment over varying batch sizes.** The gap between the linear probe accuracies of CL and NSCL ResNet-50 models (trained for 300 epochs) varies systematically with the batch size $(B)$ and their RSA alignment: when training with $\eta_t = \mathcal{O}(B)$, larger batch sizes tend to reduce alignment and increase the accuracy gap, whereas with $\eta_t = \mathcal{O}(B^{1/4})$ larger batch sizes tend to increase alignment and reduce the gap.

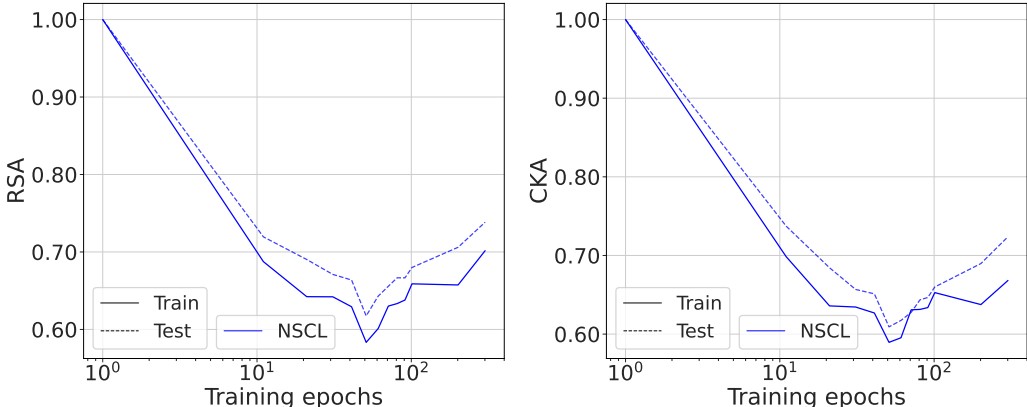

Figure 14: **CL-NSCL alignment for class-imbalanced data.** We train ResNet-50 models on SVHN (Netzer et al., 2011) with decoupled CL and NSCL objectives to analyse alignment when the classes are not uniformly distributed. The RSA and CKA values are comparable to class-balanced datasets (shown in Fig. 2- 7).

with cosine similarity trends, confirming that NSCL preserves the spatial attention structure of CL more faithfully than other supervised methods.

### B.6 Fig. 1 Methodology

We explain how to generate the plots comparing alignment in weight-space and representation-space. The two plots on the left visualize the direction of learning for each model. Each vector represents the change in model's state from initialization (epoch 0) to epoch 1000.

**Model states.** We consider CL and NSCL models trained on CIFAR100, corresponding to epoch 0 and epoch 1000-a total of four models.

**Weight space.** This plot shows how the raw parameters evolve during training. For all four models, we first flatten all the weights into a massive vector which gives us four points in a very high dimensional space (order of $10^7$). To visualize these points, we perform Principal Component Analysis (PCA) on all four vectors combined and fit them to a 3D space. This creates a shared 3D

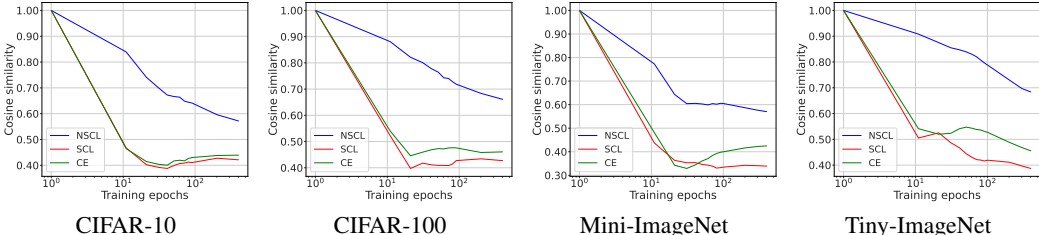

Figure 15: **Alignment in attention maps.** We evaluate cosine similarity between attention maps of decoupled CL and supervised models, and observe similar trends as for RSA/CKA values. NSCL remains the most aligned hinting at a deeper structural similarity between representations of CL and NSCL models.

coordinate system. We transform all four points into this space and we get $p_{\text{CL}}^0, p_{\text{CL}}^{1000}, p_{\text{NSCL}}^0, p_{\text{NSCL}}^{1000}$. Using these points, we create two vectors: $(v_{\text{CL}}, v_{\text{NSCL}})$, and create polar plot using the final vectors and the calculated angle between them ($85.7°$).

**Representation space.** This plot shows how model's alignment for a specific class evolved. We pick one class from our dataset (CIFAR100) and randomly sample 100 images. We use the same samples for all four models to extract their corresponding features, say $Z \in \mathbb{R}^{100 \times d}$, where $d$ is the projection dimension. We concatenate total 400 representations (100 from each model) and perform PCA to learn a shared 3D coordinate system. The representations are transformed to this shared space ($\mathbb{R}^{100 \times d} \to \mathbb{R}^{100 \times 3}$) and averaged to a single 3D point for each model. Just like before, a polar plot is created using the vectors and angle between them ($27.8°$).

**Similarity metrics.** We report RSA and CKA values computed between DCL and NSCL models trained on CIFAR100. Additionally, we show their average weight gap as detailed in Sec. 5.1. It is evident that models stay aligned in representation space but diverge in weight space.

## B.7    MODEL MERGING

In addition to our main analysis, we also conduct a simple experiment that merges models directly in representation space. Specifically, we interpolate between the learned embeddings of a CL encoder trained on the full dataset and an NSCL encoder trained on only 30% of the dataset. This merged representation already surpasses both the full-data CL model and the small-data NSCL model, reinforcing that NSCL and CL remain geometrically compatible in practice.

Concretely, given an input $x$, let $f_{\text{CL}}(x)$ and $f_{\text{NSCL}_{30}}(x)$ denote the representations from the CL encoder and the NSCL encoder trained on 30% of the dataset, respectively. We merge them via simple linear interpolation:

$$f_{\text{merged}}(x) = \alpha f_{\text{CL}}(x) + (1 - \alpha) f_{\text{NSCL}_{30}}(x).$$

We then perform NCCC and LP evaluations using the same 30% subset from the training split and report accuracy on the full mini-ImageNet test split in Fig. 17.

As shown in the figure, for all values of $\alpha$ the merged model outperforms the NSCL baseline, and for $\alpha \in [0.7, 1)$ it also outperforms the CL baseline on the mini-ImageNet downstream classification task. This suggests that the CL and NSCL representations are well aligned, making it possible to effectively merge them directly in representation space.

## C    PARAMETER-SPACE COUPLING

To complement the analysis in Sec. 4, we compare the two trajectories in parameter space. Let $e_t = \|w_t^{\text{CL}} - w_t^{\text{NSCL}}\|$ denote the parameter drift at step $t$. We would like to bound it as a function of the number of training iterations, batch size, and learning rate scheduling. We use classic techniques that can be found at (Bousquet & Elisseeff, 2002; Hardt et al., 2016; Mou et al., 2018; Kuzborskij & Lampert, 2017).

Input      DCL      NSCL      SCL      CE

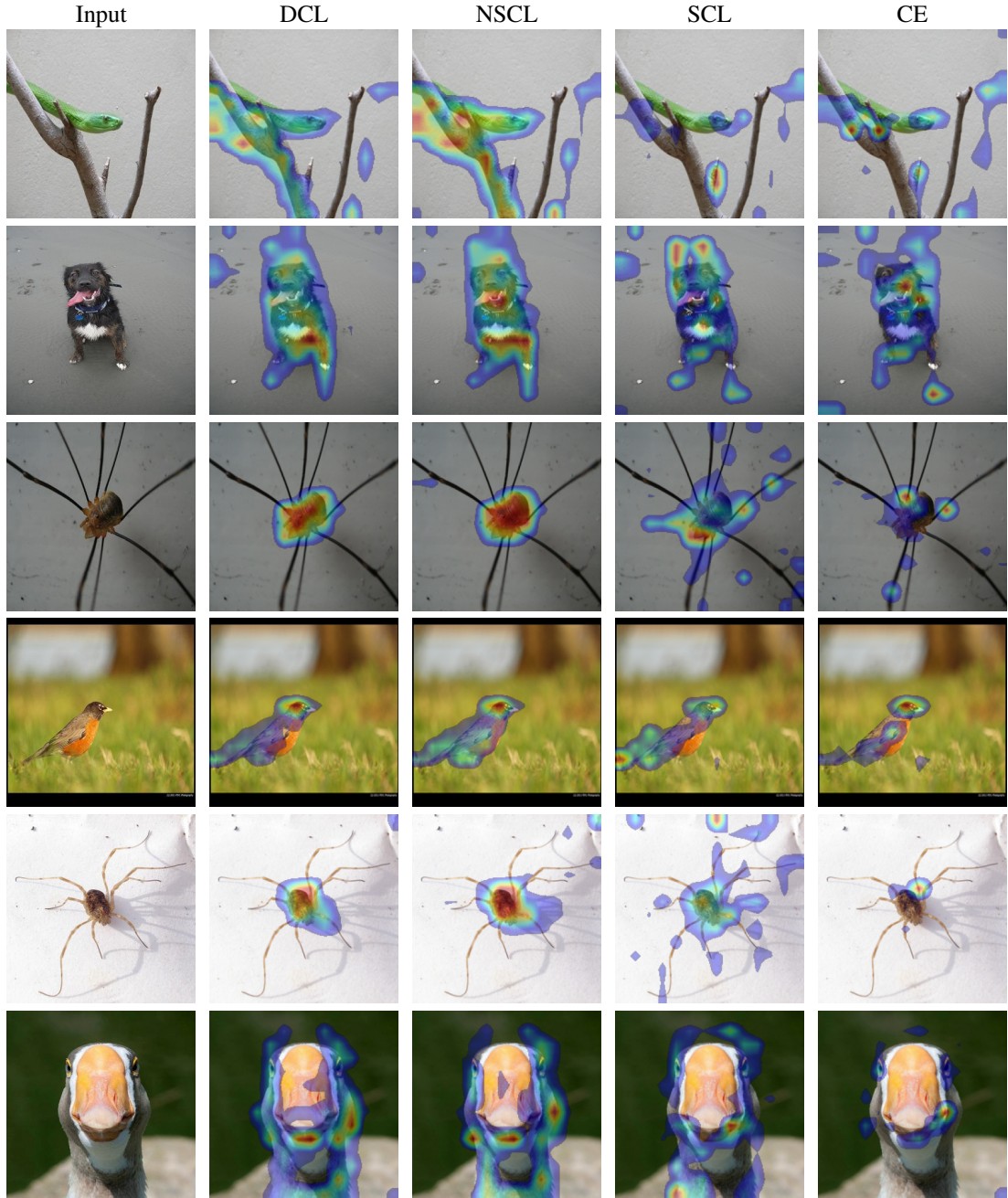

Figure 16: **Visualization of attention maps.** We visualize the self-attention of the [CLS] token from the last layer of the frozen ViT encoder. Beyond a high cosine similarity between attention maps, these visualizations reveal strong structural similarity between CL and NSCL.

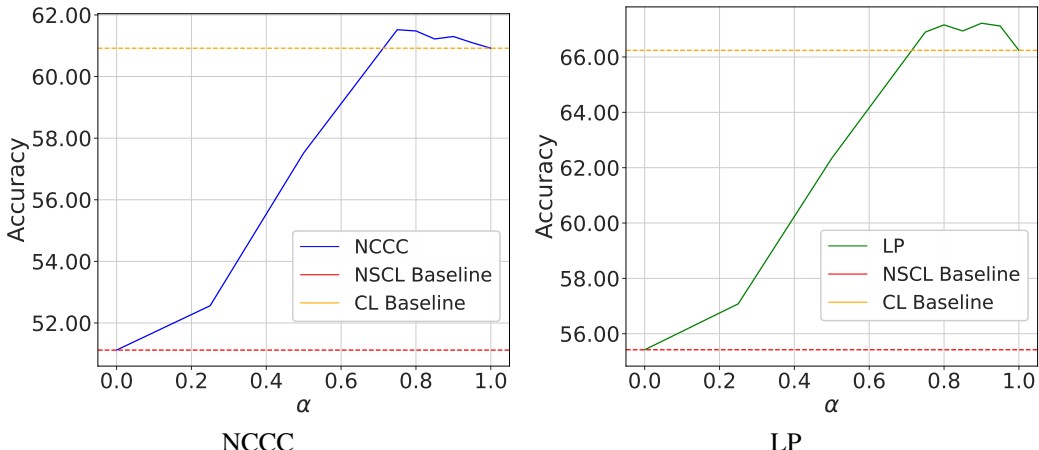

Figure 17: **Model merging in representation space:** We report NCCC and LP scores on mini-ImageNet using CL encoder trained on full dataset and NSCL encoder trained on 30% of the dataset. The performance gains obtained using merged representations illustrate the compatibility of CL and NSCL models and further support our main finding that CL and NSCL maintain closely aligned embedding geometries throughout training.

**Optimization.** In order to isolate the effect of the loss, we optimize both objectives (CL and NSCL) with standard mini-batch SGD under a single coupled protocol: at step $t$ we draw a batch $\mathcal{B}_t = \{(x_j, x_j', y_j)\}_{j=1}^B$ with replacement, where each $x_j' \sim \alpha(x_j)$ (e.g., random crop/resize, horizontal flip, color jitter, Gaussian blur); we average per-anchor terms to form either $\bar{\ell}_{\mathcal{B}_t}^{\mathrm{CL}}(w)$ or $\bar{\ell}_{\mathcal{B}_t}^{\mathrm{NS}}(w)$ using cosine similarity (optionally temperature-scaled), hence bounded in $[-1, 1]$; and we update $w_{t+1} = w_t - \eta_t \nabla \bar{\ell}_{\mathcal{B}_t}(w_t)$ with prescribed $\eta_t > 0$. We then run two coupled SGD trajectories from the same initialization $w_0^{\mathrm{CL}} = w_0^{\mathrm{NSCL}}$ that share the *same* batches and augmentations $(\mathcal{B}_t)_{t=0}^{T-1}$ and differ only by NSCL's exclusion of same-class negatives:

$$w_{t+1}^{\mathrm{CL}} = w_t^{\mathrm{CL}} - \eta_t \nabla \bar{\ell}_{\mathcal{B}_t}^{\mathrm{CL}}(w_t^{\mathrm{CL}}), \qquad w_{t+1}^{\mathrm{NSCL}} = w_t^{\mathrm{NSCL}} - \eta_t \nabla \bar{\ell}_{\mathcal{B}_t}^{\mathrm{NSCL}}(w_t^{\mathrm{NSCL}}), \quad t = 0, \ldots, T-1.$$

Throughout the analysis, we make standard assumptions on the smoothness of the loss functions and the scale of gradients.

**Assumptions.** To control the dynamics, we impose two standard conditions on the geometry of the batch objectives and the scale of pairwise gradients.

**Assumption 1** (Uniform smoothness). *For every batch $\mathcal{B}$, the functions $w \mapsto \bar{\ell}_{\mathcal{B}}^{\mathrm{CL}}(w)$ and $w \mapsto \bar{\ell}_{\mathcal{B}}^{\mathrm{NSCL}}(w)$ are $\beta$-smooth with the same constant $\beta > 0$:*

$$\|\nabla \phi(w) - \nabla \phi(v)\| \leq \beta \|w - v\| \quad \text{for all } v, w \in \mathbb{R}^p, \ \phi \in \{\bar{\ell}_{\mathcal{B}}^{\mathrm{CL}}, \bar{\ell}_{\mathcal{B}}^{\mathrm{NSCL}}\}.$$

**Assumption 2** (Bounded pairwise gradients). *There exists $G > 0$, independent of $\mathcal{B}$ and $t$, such that for all $w$ and all pairs $(u, v)$ appearing in any denominator term,*

$$\|\nabla_w \mathrm{sim}(f_w(u), f_w(v))\| \leq G.$$

We quantify drift between the coupled trajectories under shared randomness in the *nonconvex $\beta$-smooth* regime. Throughout, the only data-dependent term is $\Delta_{\pi,\delta}(B; \tau)$, which decreases with more classes and larger batches.

**Theorem 2.** *Fix $B, T \in \mathbb{N}$, $\delta \in (0, 1)$, and temperature $\tau > 0$. Suppose Assumptions 1–2 hold. Then, with probability at least $1 - \delta$,*

$$e_T \leq \frac{G}{\beta \tau} \Delta_{\pi,\delta}(B; \tau) \left( \exp\left( \beta \sum_{t=0}^{T-1} \eta_t \right) - 1 \right).$$

The bound scales linearly with $G$ and $\Delta_{\pi,\delta}(B; \tau)$, but crucially it is amplified by the exponential factor $\exp(\beta \sum_t \eta_t)$. Unless the step sizes are aggressively annealed, this term grows rapidly with

training time. Even though $\Delta_{\pi,\delta}(B;\tau)$ improves with $C$ and $B$ (e.g., for $C{=}1000$, $B{=}512$, $\delta{=}0.01$, we obtain $\Delta_{\pi,\delta}(B;\tau) \approx 0.01$ so that the reweightings of the steps differ by about one percent), the exponential accumulation can still overwhelm this small per-step gap.

In other words, parameter-space coupling guarantees only that the two runs do not drift apart too quickly in weight space. But because the weights may follow very different trajectories even when representations remain similar, this control is too weak to yield meaningful statements about representational alignment. This motivates our next step: shifting the analysis to *similarity space*, where we can obtain bounds that remain stable throughout training and translate directly into guarantees on metrics such as CKA and RSA.

**Proof idea.** With high probability over batches (Cor. 3), every anchor's denominator is dominated by negatives up to $\epsilon_{B,\delta}$ fluctuations. This keeps the (temperature–$\tau$) softmax reweighting gap between CL and NSCL small. In particular, Lem. 7 shows that the per-batch parameter gradients differ uniformly as

$$\left\| \nabla \bar{\ell}^{\mathrm{CL}}_{\mathcal{B}_t}(w) - \nabla \bar{\ell}^{\mathrm{NSCL}}_{\mathcal{B}_t}(w) \right\| \leq \frac{G}{\tau} \Delta_{\pi,\delta}(B;\tau).$$

By $\beta$-smoothness of each batch loss, each step can expand distances by at most a factor $(1 + \beta\eta_t)$. Combining this smoothness expansion with the uniform gradient-gap bound yields the following recurrence:

$$e_{t+1} \leq (1 + \beta\eta_t)\, e_t \;+\; \eta_t\, \frac{G}{\tau} \Delta_{\pi,\delta}(B;\tau),$$

where the first term propagates the previous error (with amplification controlled by curvature), and the second injects the new discrepancy introduced by the CL–NSCL gap at temperature $\tau$.

Unrolling over $T$ steps and applying the discrete Grönwall inequality gives the exponential-type bound

$$e_T \;\leq\; \frac{G}{\beta\tau} \Delta_{\pi,\delta}(B;\tau) \left( \exp\!\Big( \beta \sum_{t=0}^{T-1} \eta_t \Big) - 1 \right).$$

Thus, cumulative drift scales with the reweighting gap and is amplified exponentially with the total step size; smaller $\tau$ tightens the softmax and increases the constants (via both $1/\tau$ and $\mathrm{e}^{2/\tau}$ inside $\Delta_{\pi,\delta}$), so keeping $\sum_t \eta_t$ moderate is especially important.

# D   WHY GRADIENT DESCENT IN SIMILARITY SPACE IS A FAITHFUL SURROGATE

We now explain why running gradient descent directly in similarity space closely tracks the dynamics induced by gradient descent in parameter space.

When parameters move from $w_t$ to $w_{t+1}$, the induced change in the similarity matrix can be approximated by a linear expansion:

$$\Sigma(w_{t+1}) - \Sigma(w_t) \approx J_t(w_{t+1} - w_t), \qquad J_t := J(w_t), \tag{3}$$

where $J(w) := \partial\Sigma/\partial w$ is the Jacobian. The error in this expansion, denoted $R_t$, is quadratic in the step size:

$$\Sigma(w_{t+1}) - \Sigma(w_t) = J_t(w_{t+1} - w_t) + R_t. \tag{4}$$

By the chain rule, the gradient in parameter space can be written as follows:

$$\nabla_w \bar{\ell}(w_t) = J_t^\top \nabla_\Sigma \bar{\ell}(\Sigma(w_t)) = J_t^\top \widehat{G}_t,$$

where $\widehat{G}_t := \nabla_\Sigma \bar{\ell}(\Sigma(w_t))$. Substituting this into the update rule gives

$$\Sigma(w_{t+1}) - \Sigma(w_t) = -\eta_t\, P_t \widehat{G}_t + R_t, \qquad P_t := J_t J_t^\top \succeq 0. \tag{5}$$

Thus, parameter descent acts like similarity descent, but with a preconditioning matrix $P_t$, plus the remainder $R_t$.

Assume there exist constants $L_\Sigma, M_\Sigma > 0$ such that

$$\|J(w)\|_{2\to 2} \;\leq\; L_\Sigma, \qquad \|\Sigma(w + \Delta w) - \Sigma(w) - J(w)\Delta w\|_F \;\leq\; \frac{M_\Sigma}{2}\|\Delta w\|_2^2.$$

Then $\|P_t\|_{2\to2} \le L_\Sigma^2$ and, with $\Delta w_t := -\eta_t \nabla_w \bar\ell(w_t)$,

$$\|R_t\|_F \;\le\; \frac{M_\Sigma}{2}\eta_t^2\|\nabla_w\bar\ell(w_t)\|_2^2 \;=:\; \frac{M_\Sigma}{2}\eta_t^2\Xi_t. \tag{6}$$

Let $\widehat\Sigma_t := \Sigma(w_t)$ be the similarity trajectory induced by parameter descent. Define $\widetilde\Sigma_t$ as the trajectory of explicit similarity descent:

$$\widetilde\Sigma_{t+1} \;=\; \widetilde\Sigma_t - \eta_t\widetilde G_t, \qquad \widetilde G_t \;:=\; \nabla_\Sigma\bar\ell(\widetilde\Sigma_t),$$

with $\widehat\Sigma_0 = \widetilde\Sigma_0$. Let $E_t := \|\widehat\Sigma_t - \widetilde\Sigma_t\|_F$ and $C_\Sigma := \sup_t \|P_t - I\|_{2\to2} \le L_\Sigma^2 + 1$. Using equation 5, adding and subtracting $-\eta_t\widehat G_t$, and applying the temperature–$\tau$ bounds equation 11 and equation 6, one obtains

$$E_{t+1} \;\le\; \Big(1 + \frac{\eta_t}{2\tau^2 B}\Big)E_t \;+\; \eta_t C_\Sigma\|\widehat G_t\|_F \;+\; \frac{M_\Sigma}{2}\eta_t^2\Xi_t. \tag{7}$$

Unrolling this recursion from $E_0 = 0$ and using $\prod_u(1 + \alpha_u) \le \exp(\sum_u \alpha_u)$ yields

$$\|\widehat\Sigma_T - \widetilde\Sigma_T\|_F \;\le\; \exp\Big(\frac{1}{2\tau^2 B}\sum_{t=0}^{T-1}\eta_t\Big)\left[C_\Sigma\sum_{t=0}^{T-1}\eta_t\|\widehat G_t\|_F \;+\; \frac{M_\Sigma}{2}\sum_{t=0}^{T-1}\eta_t^2\Xi_t\right]. \tag{8}$$

By bounding $\|\widehat G_t\|_F$ via equation 13, namely $\|\widehat G_t\|_F \le \frac{1}{\tau}\sqrt{\frac{2}{B}}$, this simplifies to

$$\|\widehat\Sigma_T - \widetilde\Sigma_T\|_F \;\le\; \exp\Big(\frac{1}{2\tau^2 B}\sum_{t=0}^{T-1}\eta_t\Big)\left[\frac{\sqrt2\,C_\Sigma}{\tau\sqrt B}\sum_{t=0}^{T-1}\eta_t \;+\; \frac{M_\Sigma}{2}\sum_{t=0}^{T-1}\eta_t^2\Xi_t\right]. \tag{9}$$

To understand when this bound is conceptually reasonable, suppose $\|\nabla_w\bar\ell(w_t)\|_2 \le G$ for all $t$, so that $\Xi_t \le G^2$. The right-hand side of equation 9 is then controlled by two quantities: the cumulative step size $\sum_t \eta_t$, which appears both inside the exponential and in the linear prefactor $(\sqrt2\,C_\Sigma/(\tau\sqrt B))\sum_t \eta_t$, and the term $\sum_t \eta_t^2$.

A simple sufficient regime is to assume that $\sum_{t=0}^{T-1}\eta_t \le c_1\tau^2 B$ and $\sum_{t=0}^{T-1}\eta_t^2 \le c_2$ for fixed constants $c_1, c_2$ independent of $T$. Under these conditions, the exponential factor is bounded by $\exp\big((1/(2\tau^2 B))\sum_t \eta_t\big) \le \exp(c_1/2)$, the linear prefactor by $(\sqrt2\,C_\Sigma/(\tau\sqrt B))\sum_t \eta_t \le \sqrt2\,C_\Sigma\,c_1\,\tau\sqrt B$ (a fixed constant for given $(\tau, B)$ and moderate $c_1$), and the quadratic remainder by $(M_\Sigma/2)\sum_t \eta_t^2\Xi_t \le (M_\Sigma/2)G^2 c_2$. In particular, when $\sum_t \eta_t/(\tau^2 B)$ and $\sum_t \eta_t^2$ are both bounded by constants independent of $T$, the bound guarantees that $\|\widehat\Sigma_T - \widetilde\Sigma_T\|_F$ remains controlled (and small whenever $C_\Sigma, M_\Sigma, G$ are moderate).

To summarize, the similarity and parameter trajectories stay close whenever the normalized cumulative step size $\sum_t \eta_t/(\tau^2 B)$ is bounded and the learning-rate schedule is sufficiently decaying so that $\sum_t \eta_t^2$ remains bounded. For a fixed learning-rate schedule, a large batch size $B$ and moderate temperature $\tau$ act as stabilizing factors via the $1/(\tau\sqrt B)$ dependence in equation 9, while very small $\tau$ or extremely large, non-decaying step sizes can make the coupling poor, as reflected by the bound.

## E  TECHNICAL TOOLS AND PROOFS

### E.1  NOTATION AND BASIC SOFTMAX FACTS

Let $S = \{(x_i, y_i)\}_{i=1}^N$ be dataset with $C$ classes (each class $c$ has $n_c$ points, with $\sum_{c=1}^C n_c = N$, and we do not assume the $n_c$ are equal). For parameters $w$, let $z_i = f_w(x_i)$ and define the bounded similarity matrix

$$\Sigma(w)_{ij} \;:=\; \mathrm{sim}\big(z_i, z_j\big) \in [-1, 1].$$

At step $t$, draw a mini-batch $\mathcal{B}_t = \{(x_{j_s}, x'_{j_s}, y_{j_s})\}_{s=1}^B$ with replacement, using independent augmentations $x'_{j_s} \sim \alpha(x_{j_s})$. For an *anchor* $i \in \{j_1, \ldots, j_B\}$, let $D_i$ be its denominator index set, and let $D_i^{\mathrm{neg}} := \{k \in D_i : y_k \ne y_i\}$ (and similarly $D_i^{\mathrm{pos}}$) denote the subset restricted to negatives (e.g., in two-view SimCLR, $D_i$ consists of all $2B$ views except the anchor itself).

Define the anchor's logit vector $s_i(w) := \big(\Sigma(w)_{i,k}\big)_{k \in D_i}$ and the corresponding softmax distributions with temperature $\tau > 0$ (default 1):

$$p_i = \mathrm{softmax}\big(s_i(w)/\tau\big), \qquad q_i = \mathrm{softmax}\big((s_i(w))_{D_i^{\mathrm{neg}}}/\tau\big).$$

Let $i'$ denote the positive (augmented) index for anchor $i$.

For contrastive learning (CL) and negatives-only supervised contrastive learning (NSCL), the per-anchor and batch losses are

$$\ell_i^{\mathrm{CL}}(s_i) \;=\; -\log p_{i,i'}, \qquad \ell_i^{\mathrm{NSCL}}(s_i) \;=\; -\log q_{i,i'},$$

$$\bar{\ell}_{\mathcal{B}_t}^{\mathrm{CL}} \;=\; \frac{1}{B} \sum_{i \in \{j_1,\dots,j_B\}} \ell_i^{\mathrm{CL}}(s_i), \qquad \bar{\ell}_{\mathcal{B}_t}^{\mathrm{NSCL}} \;=\; \frac{1}{B} \sum_{i \in \{j_1,\dots,j_B\}} \ell_i^{\mathrm{NSCL}}(s_i).$$

Since $\Sigma(w)_{ij} \in [-1,1]$, each exponential term inside the softmax lies in

$$\exp\big(\Sigma(w)_{ij}/\tau\big) \in [\mathrm{e}^{-1/\tau}, \mathrm{e}^{1/\tau}],$$

a fact used below to control softmax mass ratios.

**Lemma 1** (Anchor-block orthogonality). *Fix a step $t$ and batch $\mathcal{B}_t$. For each anchor $i \in \mathcal{B}_t$, let $D_i$ be the set of indices appearing in $i$'s denominator and define the per-anchor gradient $g_i \in \mathbb{R}^{\mathcal{I}_t}$ by*

$$g_i \;:=\; \nabla_{s_i}\ell_i \quad \text{placed on the coordinates } \{(i,k) : k \in D_i\} \subset \mathcal{I}_t,$$

*with zeros elsewhere (here $\mathcal{I}_t$ is the set of all coordinates touched at step $t$). If $i \neq j$, then $g_i$ and $g_j$ have disjoint supports, and hence*

$$\langle g_i, g_j \rangle_F \;=\; 0.$$

*Consequently, for the batch gradient $G = \frac{1}{B} \sum_{i \in \mathcal{B}_t} g_i$,*

$$\|G\|_F^2 \;=\; \frac{1}{B^2} \sum_{i \in \mathcal{B}_t} \|g_i\|_F^2. \tag{10}$$

*Proof.* By construction, $g_i$ is supported only on coordinates $\{(i,k) : k \in D_i\}$, while $g_j$ is supported only on $\{(j,k) : k \in D_j\}$. For $i \neq j$ these sets are disjoint, so every coordinatewise product is zero, yielding $\langle g_i, g_j \rangle_F = 0$. Expanding the square for $G$,

$$\|G\|_F^2 \;=\; \Big\langle \frac{1}{B} \sum_i g_i, \, \frac{1}{B} \sum_j g_j \Big\rangle_F \;=\; \frac{1}{B^2} \sum_i \|g_i\|_F^2 + \frac{1}{B^2} \sum_{i \neq j} \langle g_i, g_j \rangle_F \;=\; \frac{1}{B^2} \sum_i \|g_i\|_F^2,$$

where the cross terms vanish by orthogonality. $\qquad\square$

**Lemma 2** (Softmax Hessian and gradient Lipschitzness). *Fix a step $t$ and batch $\mathcal{B}_t$. Let $\mathcal{I}_t$ be the set of coordinates $(i,k)$ that appear in any anchor's denominator at step $t$, and view $\bar{\ell}_{\mathcal{B}_t}$ (either CL or NSCL) as a function of the restricted similarity entries $\Sigma \in \mathbb{R}^{\mathcal{I}_t}$. For each anchor $i$, write $s_i = \{\Sigma(i,k) : (i,k) \in \mathcal{I}_t\}$ and $p_i = \mathrm{softmax}(s_i/\tau)$. Then:*

$$\nabla_{s_i}^2 \ell_i(s_i) \;=\; \frac{1}{\tau^2} J(s_i), \quad J(s_i) := \mathrm{Diag}(p_i) - p_i p_i^\top, \qquad \big\|\nabla^2 \bar{\ell}_{\mathcal{B}_t}(\Sigma)\big\|_{2\to2} \;\le\; \frac{1}{2\tau^2 B}.$$

*Consequently, for all $\Sigma, \widetilde{\Sigma} \in \mathbb{R}^{\mathcal{I}_t}$,*

$$\big\|\nabla_\Sigma \bar{\ell}_{\mathcal{B}_t}(\Sigma) - \nabla_\Sigma \bar{\ell}_{\mathcal{B}_t}(\widetilde{\Sigma})\big\|_F \;\le\; \frac{1}{2\tau^2 B} \big\|\Sigma - \widetilde{\Sigma}\big\|_F. \tag{11}$$

*Proof.* With temperature $\tau > 0$, for an anchor $i$ we have $p_i = \mathrm{softmax}(s_i/\tau)$ and

$$\nabla_{s_i}\ell_i(s_i) \;=\; \frac{1}{\tau}(p_i - e_{i'}) \quad\Longrightarrow\quad \nabla_{s_i}^2 \ell_i(s_i) \;=\; \frac{1}{\tau^2} \nabla_{s_i} p_i \;=\; \frac{1}{\tau^2} J(s_i),$$

where $J(s_i) := \mathrm{Diag}(p_i) - p_i p_i^\top$. Bound $\|J(s_i)\|_{2 \to 2}$ via the infinity norm:

$$
\begin{aligned}
\|J(s_i)\|_{2 \to 2} &\leq \|J(s_i)\|_\infty \\
&= \max_r \sum_\ell |J_{r\ell}| \\
&= \max_r \Big( p_{i,r}(1 - p_{i,r}) + \sum_{\ell \neq r} p_{i,r} p_{i,\ell} \Big) \\
&= \max_r 2 p_{i,r}(1 - p_{i,r}) \leq \tfrac{1}{2},
\end{aligned}
$$

since $x(1 - x) \leq 1/4$ for $x \in [0, 1]$.

The batch loss is an average over anchors, so its Hessian is block-diagonal across anchors with a prefactor $1/B$:

$$
\nabla^2 \bar{\ell}_{\mathcal{B}_t}(\Sigma) = \frac{1}{B} \mathrm{blkdiag}\Big( \tfrac{1}{\tau^2} J(s_i) \Big)_{i \in \mathcal{B}_t} = \frac{1}{\tau^2 B} \mathrm{blkdiag}\big( J(s_i) \big)_{i \in \mathcal{B}_t}.
$$

Hence

$$
\big\| \nabla^2 \bar{\ell}_{\mathcal{B}_t}(\Sigma) \big\|_{2 \to 2} = \frac{1}{\tau^2 B} \max_i \|J(s_i)\|_{2 \to 2} \leq \frac{1}{2\tau^2 B}.
$$

By the mean-value (integral) form for vector fields,

$$
\nabla_\Sigma \bar{\ell}_{\mathcal{B}_t}(\Sigma) - \nabla_\Sigma \bar{\ell}_{\mathcal{B}_t}(\widetilde{\Sigma}) = \int_0^1 \nabla^2 \bar{\ell}_{\mathcal{B}_t}\big( \widetilde{\Sigma} + \theta(\Sigma - \widetilde{\Sigma}) \big) [\Sigma - \widetilde{\Sigma}] \, d\theta,
$$

and therefore

$$
\big\| \nabla_\Sigma \bar{\ell}_{\mathcal{B}_t}(\Sigma) - \nabla_\Sigma \bar{\ell}_{\mathcal{B}_t}(\widetilde{\Sigma}) \big\|_F \leq \sup_{\theta \in [0,1]} \big\| \nabla^2 \bar{\ell}_{\mathcal{B}_t}(\Sigma_\theta) \big\|_{2 \to 2} \|\Sigma - \widetilde{\Sigma}\|_F \leq \frac{1}{2\tau^2 B} \|\Sigma - \widetilde{\Sigma}\|_F,
$$

as claimed. $\qquad \square$

**Lemma 3** (Per-anchor gradient norm and batch average). *For an anchor $i$, let $s_i$ be the vector of logits in its denominator and $p_i = \mathrm{softmax}(s_i/\tau)$. Let $i'$ denote the (unique) positive index (for NSCL, if $i'$ is not in the denominator, set $p_{i,i'} := 0$ in the display below). Then*

$$
\|\nabla_{s_i} \ell_i\|_2^2 = \frac{1}{\tau^2} \Big[ (1 - p_{i,i'})^2 + \sum_{k \neq i'} p_{i,k}^2 \Big] \leq \frac{2}{\tau^2}, \tag{12}
$$

*hence $\|\nabla_{s_i} \ell_i\|_2 \leq \sqrt{2}/\tau$. Moreover, by block orthogonality across anchors,*

$$
\Big\| \frac{1}{B} \sum_{i \in \mathcal{B}_t} \nabla_{s_i} \ell_i \Big\|_F^2 = \frac{1}{B^2} \sum_{i \in \mathcal{B}_t} \|\nabla_{s_i} \ell_i\|_2^2 \leq \frac{2}{\tau^2 B} \quad \Longrightarrow \quad \Big\| \frac{1}{B} \sum_{i \in \mathcal{B}_t} \nabla_{s_i} \ell_i \Big\|_F \leq \frac{1}{\tau} \sqrt{\frac{2}{B}}. \tag{13}
$$

*Proof.* For CL, the loss is $-\log p_{i,i'}$ with $p_i = \mathrm{softmax}(s_i/\tau)$. By the standard softmax–cross-entropy derivative with temperature,

$$
\nabla_{s_i} \ell_i = \frac{1}{\tau}(p_i - e_{i'}),
$$

so

$$
\|\nabla_{s_i} \ell_i\|_2^2 = \frac{1}{\tau^2} \Big[ (1 - p_{i,i'})^2 + \sum_{k \neq i'} p_{i,k}^2 \Big] \leq \frac{1}{\tau^2} \Big[ (1 - p_{i,i'})^2 + \Big( \sum_{k \neq i'} p_{i,k} \Big)^2 \Big] = \frac{2}{\tau^2}(1 - p_{i,i'})^2 \leq \frac{2}{\tau^2},
$$

since $p_i$ is a probability vector and $\sum_{k \neq i'} p_{i,k} = 1 - p_{i,i'}$.

For NSCL, two cases. If $i' \in D_i$, the same computation applies (the target index is present), hence the same bound holds. If $i' \notin D_i$ (negatives-only denominator), then the loss is $-\log q_{i,i'}$ with $q_i = \mathrm{softmax}\big( (s_i)_{D_i^\neg}/\tau \big)$ supported only on $D_i^\neg$, and

$$
\nabla_{s_i} \ell_i = \frac{1}{\tau} q_i \quad \text{on } D_i^{\mathrm{neg}} \quad \text{(and 0 on } D_i^{\mathrm{pos}}\text{)},
$$

so

$$\|\nabla_{s_i}\ell_i\|_2^2 \;=\; \frac{1}{\tau^2}\sum_{j\in D_i^-} q_{i,j}^2 \;\leq\; \frac{1}{\tau^2}\Big(\sum_{j\in D_i^-} q_{i,j}\Big)^2 \;=\; \frac{1}{\tau^2} \;\leq\; \frac{2}{\tau^2}.$$

Thus in all cases $\|\nabla_{s_i}\ell_i\|_2 \leq \sqrt{2}/\tau$, establishing equation 12.

For the batch bound equation 13, gradients from different anchors have disjoint supports over coordinates $\{(i,k): k\in D_i\}$, so they are orthogonal in Frobenius inner product (Lem. 1). Therefore,

$$\Big\|\frac{1}{B}\sum_{i\in\mathcal{B}_t}\nabla_{s_i}\ell_i\Big\|_F^2 = \frac{1}{B^2}\sum_{i\in\mathcal{B}_t}\|\nabla_{s_i}\ell_i\|_2^2 \leq \frac{1}{B^2}\cdot B\cdot\frac{2}{\tau^2} = \frac{2}{\tau^2 B},$$

which also implies $\big\|\frac{1}{B}\sum_{i\in\mathcal{B}_t}\nabla_{s_i}\ell_i\big\|_F \leq \frac{1}{\tau}\sqrt{2/B}$. $\square$

**Lemma 4** (Bounded logits imply bounded softmax masses). *Fix a step $t$ and an anchor $i$. Suppose all active logits satisfy $\Sigma(i,k)\in[-1,1]$. For any index subset $S$ in the anchor's denominator, define*

$$Z_S := \sum_{k\in S}\exp\big(\Sigma(i,k)/\tau\big) \quad \text{with temperature } \tau > 0.$$

*Then*

$$|S|\,\mathrm{e}^{-1/\tau} \;\leq\; Z_S \;\leq\; |S|\,\mathrm{e}^{1/\tau}.$$

*In particular, if $S_{\mathrm{pos}}$ and $S_{\mathrm{neg}}$ are the positive and negative index sets with sizes $n_{\mathrm{pos}}$ and $n_{\mathrm{neg}}$, and $Z_{\mathrm{pos}} := Z_{S_{\mathrm{pos}}}, Z_{\mathrm{neg}} := Z_{S_{\mathrm{neg}}}$, then*

$$n_{\mathrm{pos}}\,\mathrm{e}^{-1/\tau} \;\leq\; Z_{\mathrm{pos}} \;\leq\; n_{\mathrm{pos}}\,\mathrm{e}^{1/\tau}, \qquad n_{\mathrm{neg}}\,\mathrm{e}^{-1/\tau} \;\leq\; Z_{\mathrm{neg}} \;\leq\; n_{\mathrm{neg}}\,\mathrm{e}^{1/\tau},$$

*and hence*

$$\frac{Z_{\mathrm{pos}}}{Z_{\mathrm{neg}}} \;\leq\; \mathrm{e}^{2/\tau}\,\frac{n_{\mathrm{pos}}}{n_{\mathrm{neg}}} \quad \text{and} \quad \frac{Z_{\mathrm{pos}}}{Z_{\mathrm{neg}}} \;\geq\; \mathrm{e}^{-2/\tau}\,\frac{n_{\mathrm{pos}}}{n_{\mathrm{neg}}}.$$

*Proof.* Since $\Sigma(i,k)\in[-1,1]$, we have $\exp(\Sigma(i,k)/\tau)\in[\mathrm{e}^{-1/\tau},\mathrm{e}^{1/\tau}]$ for every active $k$. Summing over $k\in S$ yields $|S|\,\mathrm{e}^{-1/\tau}\leq Z_S\leq |S|\,\mathrm{e}^{1/\tau}$. Apply this with $S=S_{\mathrm{pos}}$ and $S=S_{\mathrm{neg}}$ and take ratios to obtain the stated bounds. $\square$

### E.2 HIGH-PROBABILITY BATCH COMPOSITION

Fix $T,B\in\mathbb{N}$ and $\epsilon>0$. For step $t$ and anchor $i\in\mathcal{B}_t$, let $Y_{t,s}^{(i)} = \mathbf{1}\{y_{j_s}\neq y_i\}$ for $s=1,\dots,B$.

**Lemma 5** (Batch-composition event). *For a population with $C$ classes and class priors $\pi_c = n_c/N$, the $Y_{t,s}^{(i)}$ are i.i.d. Bernoulli with mean $1-\pi_{y_i}$. For any $\epsilon>0$,*

$$\mathbb{P}\left[\exists(t,i):\; \frac{1}{B}\sum_{s=1}^{B} Y_{t,s}^{(i)} \;<\; 1-\pi_{y_i}-\epsilon\right] \;\leq\; TB\,\mathrm{e}^{-2B\epsilon^2}.$$

*Equivalently, with probability $\geq 1-TB\,\mathrm{e}^{-2B\epsilon^2}$, every anchor sees at least $B(1-\pi_{y_i}-\epsilon)$ negatives.*

*Proof.* Fix any step $t$ and anchor $i$. Because batches are drawn with replacement from a population with class priors $\pi_c = n_c/N$, for each position $s\in\{1,\dots,B\}$ the indicator $Y_{t,s}^{(i)} = \mathbf{1}\{y_{j_s}\neq y_i\}$ is Bernoulli with mean $\mathbb{E}[Y_{t,s}^{(i)}]=1-\pi_{y_i}$, and $\{Y_{t,s}^{(i)}\}_{s=1}^{B}$ are i.i.d. across $s$. By Hoeffding's inequality, for any $\epsilon>0$,

$$\mathbb{P}\left[\frac{1}{B}\sum_{s=1}^{B} Y_{t,s}^{(i)} \;<\; 1-\pi_{y_i}-\epsilon\right] = \mathbb{P}\left[\frac{1}{B}\sum_{s=1}^{B}\big(Y_{t,s}^{(i)}-\mathbb{E}Y_{t,s}^{(i)}\big) \;<\; -\epsilon\right] \;\leq\; \exp(-2B\epsilon^2).$$

There are at most $TB$ anchor–step pairs $(t,i)$ over $t=0,\dots,T-1$ and $i\in\mathcal{B}_t$. A union bound gives

$$\mathbb{P}\left[\exists(t,i):\; \frac{1}{B}\sum_{s=1}^{B} Y_{t,s}^{(i)} \;<\; 1-\pi_{y_i}-\epsilon\right] \;\leq\; TB\,\mathrm{e}^{-2B\epsilon^2}.$$

Equivalently, with probability at least $1-TB\,\mathrm{e}^{-2B\epsilon^2}$, every anchor in every step has at least $B(1-\pi_{y_i}-\epsilon)$ negatives in its denominator. $\square$

**Corollary 3.** *For $\delta \in (0,1)$, set $\epsilon_{B,\delta} := \sqrt{\frac{1}{2B}\log(\frac{TB}{\delta})}$ and let $\pi_c = n_c/N$ be the class priors and $\pi_{\max} := \max_{c\in[C]} \pi_c$. With probability $\geq 1 - \delta$, every anchor $i$ has at least $B(1 - \pi_{y_i} - \epsilon_{B,\delta})$ negatives and at most $B(\pi_{y_i} + \epsilon_{B,\delta})$ positives in its denominator. In particular,*

$$|D_i^{\mathrm{neg}}| \geq B\big(1 - \pi_{\max} - \epsilon_{B,\delta}\big), \qquad |D_i^{\mathrm{pos}}| \leq B\big(\pi_{\max} + \epsilon_{B,\delta}\big).$$

*Using bounded logits, the ratio of total positive to negative softmax mass (at temperature $\tau > 0$) satisfies, for all anchors and steps,*

$$\frac{Z_i^{\mathrm{pos}}}{Z_i^{\mathrm{neg}}} \leq \frac{\mathrm{e}^{2/\tau}\big(\pi_{\max} + \epsilon_{B,\delta}\big)}{1 - \pi_{\max} - \epsilon_{B,\delta}} = \tfrac{1}{2}\Delta_{\pi,\delta}(B;\tau), \qquad (14)$$

*where*

$$\Delta_{\pi,\delta}(B;\tau) = \frac{2\,\mathrm{e}^{2/\tau}\big(\pi_{\max} + \epsilon_{B,\delta}\big)}{1 - \pi_{\max} - \epsilon_{B,\delta}}.$$

*Proof.* Set $\epsilon = \epsilon_{B,\delta} := \sqrt{\frac{1}{2B}\log(\frac{TB}{\delta})}$ and $\Delta_{\pi,\delta}(B;\tau) := \frac{2\,\mathrm{e}^{2/\tau}\big(\pi_{\max}+\epsilon_{B,\delta}\big)}{1-\pi_{\max}-\epsilon_{B,\delta}}$. Apply Lem. 5 with this $\epsilon$: with probability at least $1 - \delta$, for every step $t$ and every anchor $i$,

$$|D_i^{\mathrm{neg}}| \geq B\big(1 - \pi_{y_i} - \epsilon_{B,\delta}\big), \qquad |D_i^{\mathrm{pos}}| \leq B\big(\pi_{y_i} + \epsilon_{B,\delta}\big).$$

In particular,

$$|D_i^{\mathrm{neg}}| \geq B\big(1 - \pi_{\max} - \epsilon_{B,\delta}\big), \qquad |D_i^{\mathrm{pos}}| \leq B\big(\pi_{\max} + \epsilon_{B,\delta}\big).$$

In two-view SimCLR, each sampled point contributes two denominator entries, so the denominator contains at least $2|D_i^{\mathrm{neg}}|$ negative entries and at most $2|D_i^{\mathrm{pos}}|$ positive entries; the factor 2 cancels in the ratio below.

Because similarities are bounded in $[-1,1]$, each logit lies in $[-1,1]$ and hence each exponential term at temperature $\tau$ lies in $[\mathrm{e}^{-1/\tau}, \mathrm{e}^{1/\tau}]$. Therefore, for any anchor and step,

$$Z_i^{\mathrm{pos}} \leq \mathrm{e}^{1/\tau} \cdot (2|D_i^{\mathrm{pos}}|), \qquad Z_i^{\mathrm{neg}} \geq \mathrm{e}^{-1/\tau} \cdot (2|D_i^{\mathrm{neg}}|),$$

and thus

$$\frac{Z_i^{\mathrm{pos}}}{Z_i^{\mathrm{neg}}} \leq \mathrm{e}^{2/\tau}\frac{|D_i^{\mathrm{pos}}|}{|D_i^{\mathrm{neg}}|} \leq \frac{\mathrm{e}^{2/\tau}\big(\pi_{\max} + \epsilon_{B,\delta}\big)}{1 - \pi_{\max} - \epsilon_{B,\delta}} = \tfrac{1}{2}\Delta_{\pi,\delta}(B;\tau).$$

The bound is meaningful whenever $\epsilon_{B,\delta} < 1 - \pi_{\max}$ so that the denominator is positive. This proves the corollary. $\square$

**Lemma 6** (Per-anchor reweighting gap). *On the event of Cor. 3, let $p$ be the CL softmax (temperature $\tau > 0$) over an anchor's full denominator, and $q$ the NSCL softmax (same $\tau$) that removes same-class entries and renormalizes over negatives. Then*

$$\|p - q\|_1 \leq \Delta_{\pi,\delta}(B;\tau), \qquad \|p - q\|_2 \leq \|p - q\|_1 \leq \Delta_{\pi,\delta}(B;\tau).$$

*Proof.* Fix an anchor $i$ and let $D_i^{\mathrm{pos}}, D_i^{\mathrm{neg}}$ be its positive and negative index sets in the CL denominator. Write $s_k := \Sigma(i,k)$ and define

$$Z_i^{\mathrm{pos}} := \sum_{k \in D_i^{\mathrm{pos}}} \exp\big(s_k/\tau\big), \qquad Z_i^{\mathrm{neg}} := \sum_{j \in D_i^{\mathrm{neg}}} \exp\big(s_j/\tau\big), \qquad \alpha := \frac{Z_i^{\mathrm{pos}}}{Z_i^{\mathrm{pos}} + Z_i^{\mathrm{neg}}}.$$

Let $p$ be the CL softmax on $D_i^{\mathrm{pos}} \cup D_i^{\mathrm{neg}}$ and let $q$ be the NSCL softmax that zeros positive entries and renormalizes on negatives: $q(k) = 0$ for $k \in D_i^{\mathrm{pos}}$ and $q(j) = p(j)/(1-\alpha)$ for $j \in D_i^{\mathrm{neg}}$. Then

$$\|p - q\|_1 = \sum_{k \in D_i^{\mathrm{pos}}} p_k + \sum_{j \in D_i^{\mathrm{neg}}} \left| p_j - \frac{p_j}{1-\alpha} \right| = \alpha + (1-\alpha)\frac{\alpha}{1-\alpha} = 2\alpha \leq \frac{2Z_i^{\mathrm{pos}}}{Z_i^{\mathrm{neg}}}.$$

On the high-probability event of Cor. 3, since $s \in [-1,1] \Rightarrow \exp(s/\tau) \in [\mathrm{e}^{-1/\tau}, \mathrm{e}^{1/\tau}]$,

$$Z_i^{\mathrm{pos}} \leq \mathrm{e}^{1/\tau}|D_i^{\mathrm{pos}}|, \qquad Z_i^{\mathrm{neg}} \geq \mathrm{e}^{-1/\tau}|D_i^{\mathrm{neg}}|.$$

Moreover, by Cor. 3,

$$|D_i^{\text{pos}}| \leq 2B\Big(\pi_{\max} + \epsilon_{B,\delta}\Big), \qquad |D_i^{\text{neg}}| \geq 2B\Big(1 - \pi_{\max} - \epsilon_{B,\delta}\Big),$$

(each sampled point contributes two keys, so the factor 2 cancels in the ratio). Hence

$$\frac{2Z_i^{\text{pos}}}{Z_i^{\text{neg}}} \leq 2\,\text{e}^{2/\tau}\,\frac{|D_i^{\text{pos}}|}{|D_i^{\text{neg}}|} \leq \frac{2\,\text{e}^{2/\tau}\big(\pi_{\max} + \epsilon_{B,\delta}\big)}{1 - \pi_{\max} - \epsilon_{B,\delta}} = \Delta_{\pi,\delta}(B;\tau).$$

Therefore $\|p - q\|_1 \leq \Delta_{\pi,\delta}(B;\tau)$. Finally, $\|p - q\|_2 \leq \|p - q\|_1$ yields the second claim. $\qquad\square$

### E.3 PARAMETER-SPACE COUPLING: SUPPORTING LEMMAS AND PROOFS

**Lemma 7** (Per-batch parameter-gradient gap). *On the event of Cor. 3, for any step $t$ and any $w$,*

$$\big\|\nabla\bar{\ell}_{\mathcal{B}_t}^{\text{CL}}(w) - \nabla\bar{\ell}_{\mathcal{B}_t}^{\text{NSCL}}(w)\big\| \leq \frac{G}{\tau}\,\Delta_{\pi,\delta}(B;\tau).$$

*Proof.* Fix $t$ and $w$. For an anchor $i \in \mathcal{B}_t$, let $D_i$ be its denominator index set, split as $D_i = \text{pos}_i \cup \text{neg}_i$, where $\text{pos}_i$ collects all same-class indices (including the designated positive $i'$) and $\text{neg}_i$ the rest. Write the logits $s_{ik} = \Sigma(i, k)$, the CL softmax $p_{ik} = \exp(s_{ik}/\tau)/\sum_{\ell\in D_i}\exp(s_{i\ell}/\tau)$, and the NSCL softmax over negatives $q_{ij} = p_{ij}/(1 - \alpha_i)$ for $j \in \text{neg}_i$, with $q_k = 0$ for $k \in \text{pos}_i$, where $\alpha_i := \sum_{k\in\text{pos}_i} p_{ik}$. Define $v_{ik} := \nabla_w s_{ik} = \nabla_w \text{sim}\big(f_w(x_i), f_w(x_k)\big)$; by Assumption 2, $\|v_{ik}\| \leq G$ for all $(i, k)$.

For the per-anchor losses,

$$\nabla_w \ell_{i,\mathcal{B}_t}^{\text{CL}} = \frac{1}{\tau}\Big(\sum_{k\in D_i} p_{ik}\,v_{ik} - v_{ii'}\Big), \qquad \nabla_w \ell_{i,\mathcal{B}_t}^{\text{NSCL}} = \frac{1}{\tau}\Big(\sum_{j\in\text{neg}_i} q_{ij}\,v_{ij} - v_{ii'}\Big).$$

Hence the per-anchor gradient difference is

$$\Delta g_i := \nabla_w \ell_{i,\mathcal{B}_t}^{\text{CL}} - \nabla_w \ell_{i,\mathcal{B}_t}^{\text{NSCL}} = \frac{1}{\tau}\left(\underbrace{\sum_{k\in\text{pos}_i} p_{ik}\,v_{ik}}_{(A)} + \underbrace{\sum_{j\in\text{neg}_i} (p_{ij} - q_{ij})\,v_{ij}}_{(B)}\right).$$

By the triangle inequality and $\|v_{ik}\| \leq G$,

$$\|\Delta g_i\| \leq \frac{G}{\tau}\Big(\sum_{k\in\text{pos}_i} p_{ik} + \sum_{j\in\text{neg}_i} |p_{ij} - q_{ij}|\Big).$$

Since $q_{ij} = p_{ij}/(1 - \alpha_i)$ for $j \in \text{neg}_i$,

$$\sum_{j\in\text{neg}_i} |p_{ij} - q_{ij}| = \sum_{j\in\text{neg}_i} p_{ij}\,\frac{\alpha_i}{1 - \alpha_i} = \alpha_i.$$

Therefore $\|\Delta g_i\| \leq \frac{G}{\tau}(\alpha_i + \alpha_i) = \frac{2G}{\tau}\alpha_i$. Writing $r_i := \frac{Z_{\text{pos}}}{Z_{\text{neg}}}$ with $Z_{\text{pos}} = \sum_{k\in\text{pos}_i}\exp(s_{ik}/\tau)$, $Z_{\text{neg}} = \sum_{j\in\text{neg}_i}\exp(s_{ij}/\tau)$, we have $\alpha_i = \frac{r_i}{1+r_i}$, hence $2\alpha_i = \frac{2r_i}{1+r_i} \leq 2r_i$, so

$$\|\Delta g_i\| \leq \frac{2G}{\tau}\,\frac{Z_{\text{pos}}}{Z_{\text{neg}}}.$$

On the high-probability event of Cor. 3, for every anchor

$$\frac{Z_{\text{pos}}}{Z_{\text{neg}}} \leq \frac{\text{e}^{2/\tau}\big(\pi_{\max} + \epsilon_{B,\delta}\big)}{1 - \pi_{\max} - \epsilon_{B,\delta}} = \tfrac{1}{2}\Delta_{\pi,\delta}(B;\tau),$$

so $\|\Delta g_i\| \leq \frac{G}{\tau}\Delta_{\pi,\delta}(B;\tau)$ for all anchors $i$.

Finally, the batch gradients are averages over anchors:

$$\nabla \bar{\ell}_{\mathcal{B}_t}^{\mathrm{CL}} - \nabla \bar{\ell}_{\mathcal{B}_t}^{\mathrm{NSCL}} = \frac{1}{B} \sum_{i \in \mathcal{B}_t} \Delta g_i,$$

hence

$$\left\| \nabla \bar{\ell}_{\mathcal{B}_t}^{\mathrm{CL}} - \nabla \bar{\ell}_{\mathcal{B}_t}^{\mathrm{NSCL}} \right\| \leq \frac{1}{B} \sum_{i \in \mathcal{B}_t} \|\Delta g_i\| \leq \frac{1}{B} \sum_{i \in \mathcal{B}_t} \frac{G}{\tau} \Delta_{\pi,\delta}(B;\tau) = \frac{G}{\tau} \Delta_{\pi,\delta}(B;\tau).$$

$\square$

**Theorem 2.** *Fix $B, T \in \mathbb{N}$, $\delta \in (0,1)$, and temperature $\tau > 0$. Suppose Assumptions 1–2 hold. Then, with probability at least $1 - \delta$,*

$$e_T \leq \frac{G}{\beta \tau} \Delta_{\pi,\delta}(B;\tau) \left( \exp\left( \beta \sum_{t=0}^{T-1} \eta_t \right) - 1 \right).$$

*Proof.* Let $\Phi_t^{\mathrm{CL}}(w) := \bar{\ell}_{\mathcal{B}_t}^{\mathrm{CL}}(w)$ and $\Phi_t^{\mathrm{NSCL}}(w) := \bar{\ell}_{\mathcal{B}_t}^{\mathrm{NSCL}}(w)$. Assume each $\Phi_t^{\mathrm{CL}}$ is $\beta$-smooth. Set $e_t := \|w_t^{\mathrm{CL}} - w_t^{\mathrm{NSCL}}\|$.

Write

$$
\begin{aligned}
e_{t+1} &= \left\| w_{t+1}^{\mathrm{CL}} - w_{t+1}^{\mathrm{NSCL}} \right\| = \left\| T_t(w_t^{\mathrm{CL}}) - \left( w_t^{\mathrm{NSCL}} - \eta_t \nabla \Phi_t^{\mathrm{NSCL}}(w_t^{\mathrm{NSCL}}) \right) \right\| \\
&\leq \underbrace{\left\| T_t(w_t^{\mathrm{CL}}) - T_t(w_t^{\mathrm{NSCL}}) \right\|}_{\mathrm{(I)}} + \eta_t \underbrace{\left\| \nabla \Phi_t^{\mathrm{CL}}(w_t^{\mathrm{NSCL}}) - \nabla \Phi_t^{\mathrm{NSCL}}(w_t^{\mathrm{NSCL}}) \right\|}_{\mathrm{(II)}}.
\end{aligned}
$$

*Bounding (I).* Using the integral Hessian representation,

$$\nabla \Phi_t^{\mathrm{CL}}(u) - \nabla \Phi_t^{\mathrm{CL}}(v) = H_t(v,u)\,(u-v), \qquad H_t(v,u) := \int_0^1 \nabla^2 \Phi_t^{\mathrm{CL}}(v + \tau(u-v))\, d\tau,$$

and $\beta$-smoothness gives $\|H_t(v,u)\|_{2\to 2} \leq \beta$. Hence

$$
\begin{aligned}
\|T_t(u) - T_t(v)\| &= \|(I - \eta_t H_t(v,u))(u-v)\| \\
&\leq \left\| I - \eta_t H_t(v,u) \right\|_{2\to 2} \|u-v\| \\
&\leq (1 + \eta_t \beta)\, \|u-v\|.
\end{aligned}
$$

Thus, (I) $\leq (1 + \eta_t \beta)\, e_t$.

*Bounding (II).* On the high-probability event of Cor. 3, Lem. 7 yields

$$\mathrm{(II)} \leq \frac{G}{\tau} \Delta_{\pi,\delta}(B;\tau).$$

Combining the bounds,

$$e_{t+1} \leq (1 + \eta_t \beta)\, e_t + \eta_t \frac{G}{\tau} \Delta_{\pi,\delta}(B;\tau). \tag{15}$$

Iterating equation 15 from $e_0 = 0$ gives

$$e_T \leq \sum_{t=0}^{T-1} \eta_t \frac{G}{\tau} \Delta_{\pi,\delta}(B;\tau) \prod_{s=t+1}^{T-1} (1 + \eta_s \beta) \leq \frac{G}{\tau} \Delta_{\pi,\delta}(B;\tau) \sum_{t=0}^{T-1} \eta_t \exp\left( \beta \sum_{s=t+1}^{T-1} \eta_s \right),$$

where we used $1 + x \leq e^x$. Let $S_k := \sum_{s=k}^{T-1} \eta_s$ so that $S_t = \eta_t + S_{t+1}$. Then for each $t$,

$$\eta_t \exp(\beta S_{t+1}) \leq \frac{1}{\beta} \left( \exp(\beta S_t) - \exp(\beta S_{t+1}) \right),$$

since $e^{\beta \eta_t} - 1 \geq \beta \eta_t$. Summing over $t = 0, \ldots, T-1$ telescopes to

$$e_T \leq \frac{G}{\beta \tau} \Delta_{\pi,\delta}(B;\tau) \left( \exp\left( \beta \sum_{t=0}^{T-1} \eta_t \right) - 1 \right).$$

This holds with probability at least $1 - \delta$. $\square$

### E.4 SIMILARITY-SPACE ANALYSIS AND COUPLING

**Lemma 8** (Per-step gradient gap in similarity space). *On the event of Cor. 3, for any step $t$,*

$$\left\|G_t^{\mathrm{CL}}(\Sigma_t^{\mathrm{CL}}) - G_t^{\mathrm{NSCL}}(\Sigma_t^{\mathrm{NSCL}})\right\|_F \;\leq\; \underbrace{\frac{1}{\tau}\cdot\frac{\Delta_{\pi,\delta}(B;\tau)}{\sqrt{B}}}_{reweighting\;(block\text{-}orth.)} \;+\; \underbrace{\frac{1}{2\tau^2 B}\left\|\Sigma_t^{\mathrm{CL}} - \Sigma_t^{\mathrm{NSCL}}\right\|_F}_{Lipschitz\;in\;\Sigma}.$$

*Proof.* Add and subtract $G_t^{\mathrm{NSCL}}(\Sigma_t^{\mathrm{CL}})$ and apply the triangle inequality:

$$\left\|G_t^{\mathrm{CL}}(\Sigma_t^{\mathrm{CL}}) - G_t^{\mathrm{NSCL}}(\Sigma_t^{\mathrm{NSCL}})\right\|_F$$
$$\leq \underbrace{\left\|G_t^{\mathrm{CL}}(\Sigma_t^{\mathrm{CL}}) - G_t^{\mathrm{NSCL}}(\Sigma_t^{\mathrm{CL}})\right\|_F}_{(A)} + \underbrace{\left\|G_t^{\mathrm{NSCL}}(\Sigma_t^{\mathrm{CL}}) - G_t^{\mathrm{NSCL}}(\Sigma_t^{\mathrm{NSCL}})\right\|_F}_{(B)}. \tag{16}$$

*Term (B): Lipschitz in $\Sigma$.* By the temperature-$\tau$ softmax–Hessian bound equation 11,

$$(B) \;\leq\; \frac{1}{2\tau^2 B}\left\|\Sigma_t^{\mathrm{CL}} - \Sigma_t^{\mathrm{NSCL}}\right\|_F.$$

*Term (A): reweighting gap at fixed $\Sigma_t^{\mathrm{CL}}$.* Decompose the batch gradient into anchor blocks:

$$G_t^{\circ}(\Sigma) \;=\; \frac{1}{B}\sum_{i\in\mathcal{B}_t} g_{t,i}^{\circ}(\Sigma), \qquad \circ \in \{\mathrm{CL}, \mathrm{NSCL}\},$$

where each $g_{t,i}^{\circ}$ has support only on the coordinates of anchor $i$. For anchor $i$, with temperature $\tau$, $g_{t,i}^{\mathrm{CL}}(\Sigma_t^{\mathrm{CL}}) = (1/\tau)(p_i - e_{i'})$, $g_{t,i}^{\mathrm{NSCL}}(\Sigma_t^{\mathrm{CL}}) = (1/\tau)(q_i - e_{i'})$, so $g_{t,i}^{\mathrm{CL}}(\Sigma_t^{\mathrm{CL}}) - g_{t,i}^{\mathrm{NSCL}}(\Sigma_t^{\mathrm{CL}}) = (1/\tau)(p_i - q_i)$ on that block. By block orthogonality (Lem. 1),

$$(A) \;=\; \frac{1}{B}\left\|\sum_{i\in\mathcal{B}_t}\frac{1}{\tau}(p_i - q_i)\right\|_F \;=\; \frac{1}{\tau B}\sqrt{\sum_{i\in\mathcal{B}_t}\|p_i - q_i\|_2^2}.$$

On the event of Cor. 3, Lem. 6 gives $\|p_i - q_i\|_2 \leq \Delta_{\pi,\delta}(B;\tau)$ for every anchor, hence

$$(A) \;\leq\; \frac{1}{\tau B}\sqrt{B\,\Delta_{\pi,\delta}(B;\tau)^2} \;=\; \frac{1}{\tau}\cdot\frac{\Delta_{\pi,\delta}(B;\tau)}{\sqrt{B}}.$$

Combining the bounds on (A) and (B) yields the claim. $\qquad\square$

**Theorem 1** (Similarity-space coupling). *Fix $B, T \in \mathbb{N}$, $\delta \in (0,1)$, and temperature $\tau > 0$. Consider the coupled similarity-descent recursions equation 1 for CL and NSCL with shared initialization and shared mini-batches/augmentations. Then, with probability at least $1 - \delta$ over the draws of the mini-batches and augmentations, for any stepsizes $(\eta_t)_{t=0}^{T-1}$,*

$$\left\|\Sigma_T^{\mathrm{CL}} - \Sigma_T^{\mathrm{NSCL}}\right\|_F \;\leq\; \exp\!\Big(\frac{1}{2\tau^2 B}\sum_{t=0}^{T-1}\eta_t\Big)\frac{1}{\tau\sqrt{B}}\Big(\sum_{t=0}^{T-1}\eta_t\Big)\Delta_{\pi,\delta}(B;\tau). \tag{2}$$

*Proof.* Condition on the event of Cor. 3 (which holds with probability at least $1 - \delta$). Let $D_t := \|\Sigma_t^{\mathrm{CL}} - \Sigma_t^{\mathrm{NSCL}}\|_F$. From the coupled updates equation 1,

$$\Sigma_{t+1}^{\mathrm{CL}} - \Sigma_{t+1}^{\mathrm{NSCL}} \;=\; \big(\Sigma_t^{\mathrm{CL}} - \Sigma_t^{\mathrm{NSCL}}\big) - \eta_t\big(G_t^{\mathrm{CL}}(\Sigma_t^{\mathrm{CL}}) - G_t^{\mathrm{NSCL}}(\Sigma_t^{\mathrm{NSCL}})\big),$$

hence

$$D_{t+1} \;\leq\; D_t + \eta_t\left\|G_t^{\mathrm{CL}}(\Sigma_t^{\mathrm{CL}}) - G_t^{\mathrm{NSCL}}(\Sigma_t^{\mathrm{NSCL}})\right\|_F.$$

Add and subtract $G_t^{\mathrm{NSCL}}(\Sigma_t^{\mathrm{CL}})$ and apply Lem. 8 (reweighting gap + Lipschitz with temperature $\tau$):

$$\left\|G_t^{\mathrm{CL}}(\Sigma_t^{\mathrm{CL}}) - G_t^{\mathrm{NSCL}}(\Sigma_t^{\mathrm{NSCL}})\right\|_F \;\leq\; \frac{1}{\tau}\cdot\frac{\Delta_{\pi,\delta}(B;\tau)}{\sqrt{B}} + \frac{1}{2\tau^2 B}D_t.$$

Therefore,

$$D_{t+1} \leq \Big(1 + \frac{\eta_t}{2\tau^2 B}\Big) D_t + \eta_t \frac{1}{\tau} \cdot \frac{\Delta_{\pi,\delta}(B;\tau)}{\sqrt{B}}.$$

Let $\alpha_t := \frac{\eta_t}{2\tau^2 B}$ and $\gamma_t := \eta_t \frac{\Delta_{\pi,\delta}(B;\tau)}{\tau\sqrt{B}}$. With $D_0 = 0$ (shared initialization), the discrete Grönwall/product form gives

$$D_T \leq \sum_{s=0}^{T-1} \gamma_s \prod_{u=s+1}^{T-1} (1 + \alpha_u) \leq \exp\Big(\sum_{u=0}^{T-1} \alpha_u\Big) \sum_{s=0}^{T-1} \gamma_s,$$

using $\prod_u (1 + \alpha_u) \leq \exp(\sum_u \alpha_u)$. Substituting $\alpha_t, \gamma_t$ yields

$$D_T \leq \exp\Big(\frac{1}{2\tau^2 B} \sum_{t=0}^{T-1} \eta_t\Big) \frac{1}{\tau\sqrt{B}} \Big(\sum_{t=0}^{T-1} \eta_t\Big) \Delta_{\pi,\delta}(B;\tau),$$

as desired. $\qquad\square$

**Consequences for CKA and RSA.**

**Corollary 1** (CKA lower bound). *In the setting of Thm. 1. Assume $\|K_T^{\mathrm{CL}}\|_F > 0$. With probability at least $1 - \delta$,*

$$\mathrm{CKA}_T \geq \frac{1 - \rho_T}{1 + \rho_T}, \qquad \rho_T \leq \frac{\exp\big(\frac{1}{2\tau^2 B} \sum_{t=0}^{T-1} \eta_t\big) \frac{1}{\tau\sqrt{B}} \big(\sum_{t=0}^{T-1} \eta_t\big) \Delta_{\pi,\delta}(B;\tau)}{\|K_T^{\mathrm{CL}}\|_F}.$$

*Proof.* Let $A_T := \|K_T^{\mathrm{CL}}\|_F > 0$ and $\Delta_{K,T} := \|K_T^{\mathrm{CL}} - K_T^{\mathrm{NSCL}}\|_F$, where all norms are Frobenius. Then

$$\begin{aligned} \langle K_T^{\mathrm{CL}}, K_T^{\mathrm{NSCL}} \rangle &= \langle K_T^{\mathrm{CL}}, K_T^{\mathrm{CL}} + (K_T^{\mathrm{NSCL}} - K_T^{\mathrm{CL}}) \rangle \\ &= \|K_T^{\mathrm{CL}}\|_F^2 + \langle K_T^{\mathrm{CL}}, K_T^{\mathrm{NSCL}} - K_T^{\mathrm{CL}} \rangle \geq A_T^2 - A_T \Delta_{K,T}, \end{aligned} \tag{17}$$

by Cauchy–Schwarz. By the triangle inequality, $\|K_T^{\mathrm{NSCL}}\|_F \leq A_T + \Delta_{K,T}$. Hence

$$\mathrm{CKA}_T = \frac{\langle K_T^{\mathrm{CL}}, K_T^{\mathrm{NSCL}} \rangle}{\|K_T^{\mathrm{CL}}\|_F \|K_T^{\mathrm{NSCL}}\|_F} \geq \frac{A_T^2 - A_T \Delta_{K,T}}{A_T(A_T + \Delta_{K,T})} = \frac{1 - \Delta_{K,T}/A_T}{1 + \Delta_{K,T}/A_T}.$$

Next, $K_T^\circ = H\Sigma_T^\circ H$ with the centering projector $H = I - \frac{1}{N}\mathbf{1}\mathbf{1}^\top$, so $\Delta_{K,T} = \|H(\Sigma_T^{\mathrm{CL}} - \Sigma_T^{\mathrm{NSCL}})H\|_F \leq \|\Sigma_T^{\mathrm{CL}} - \Sigma_T^{\mathrm{NSCL}}\|_F$ because $\|H\|_{2\to2} = 1$. By Thm. 1, with probability at least $1 - \delta$,

$$\|\Sigma_T^{\mathrm{CL}} - \Sigma_T^{\mathrm{NSCL}}\|_F \leq \exp\Big(\frac{1}{2\tau^2 B} \sum_{t=0}^{T-1} \eta_t\Big) \frac{1}{\tau\sqrt{B}} \Big(\sum_{t=0}^{T-1} \eta_t\Big) \Delta_{\pi,\delta}(B;\tau).$$

Combining the last two equations yields the lower bound on $\mathrm{CKA}_T$ with probability at least $1 - \delta$. $\qquad\square$

**Corollary 2** (RSA lower bound). *In the setting of Thm. 1. Assume $\sigma_{D,T} > 0$. With probability at least $1 - \delta$,*

$$\mathrm{RSA}_T \geq \frac{1 - r_T}{1 + r_T}, \qquad r_T \leq \frac{\exp\big(\frac{1}{2\tau^2 B} \sum_{t=0}^{T-1} \eta_t\big) \frac{1}{\tau\sqrt{B}} \big(\sum_{t=0}^{T-1} \eta_t\big) \Delta_{\pi,\delta}(B;\tau)}{\sqrt{M}\,\sigma_{D,T}}.$$

*Proof.* Let $M = \binom{N}{2}$ and let $C := I - \frac{1}{M}\mathbf{1}\mathbf{1}^\top$ be the centering projector in $\mathbb{R}^M$. Write $a_c := Ca_T$ and $b_c := Cb_T$. Then

$$\mathrm{RSA}_T = \frac{\langle a_c, b_c \rangle}{\|a_c\|_2 \|b_c\|_2}.$$

For any nonzero $u$ and any $v$ in an inner-product space,

$$\langle u, v \rangle = \langle u, u + (v - u) \rangle = \|u\|_2^2 + \langle u, v - u \rangle \geq \|u\|_2^2 - \|u\|_2 \|v - u\|_2,$$

and $\|v\|_2 \leq \|u\|_2 + \|v - u\|_2$. Therefore,

$$\frac{\langle u, v \rangle}{\|u\|_2 \|v\|_2} \geq \frac{1 - \|v - u\|_2/\|u\|_2}{1 + \|v - u\|_2/\|u\|_2}.$$

Apply this with $u = a_c$ and $v = b_c$ to obtain

$$\mathrm{RSA}_T \ \geq \ \frac{1 - \|b_c - a_c\|_2/\|a_c\|_2}{1 + \|b_c - a_c\|_2/\|a_c\|_2}.$$

Since $C$ is an orthogonal projector, $\|b_c - a_c\|_2 = \|C(b_T - a_T)\|_2 \leq \|b_T - a_T\|_2$. By construction of the RDM vectors,

$$b_T - a_T \ = \ -\mathrm{vec}\left(\mathrm{off}\left(\Sigma_T^{\mathrm{NSCL}} - \Sigma_T^{\mathrm{CL}}\right)\right),$$

so $\|b_T - a_T\|_2 = \|\mathrm{off}(\Sigma_T^{\mathrm{NSCL}} - \Sigma_T^{\mathrm{CL}})\|_F \leq \|\Sigma_T^{\mathrm{NSCL}} - \Sigma_T^{\mathrm{CL}}\|_F$. Finally, by Thm. 1, with probability at least $1 - \delta$,

$$\|\Sigma_T^{\mathrm{NSCL}} - \Sigma_T^{\mathrm{CL}}\|_F \ \leq \ \exp\left(\frac{1}{2\tau^2 B}\sum_{t=0}^{T-1}\eta_t\right)\frac{1}{\tau\sqrt{B}}\left(\sum_{t=0}^{T-1}\eta_t\right)\Delta_{\pi,\delta}(B;\tau).$$

Combining the last three displays yields the stated $(1 - r)/(1 + r)$ lower bound on $\mathrm{RSA}_T$ after substituting $\|a_c\|_2 = \sqrt{M}\,\sigma_{D,T}$. $\qquad\square$

