# OpenReview forum: "On the Alignment Between Supervised and Self-Supervised Contrastive Learning"
_ICLR.cc/2026/Conference — ICLR 2026 Poster_

### Official Review · Reviewer_nMEv · 2025-10-24

**Soundness:** 2
**Presentation:** 3
**Contribution:** 2
**Rating:** 2
**Confidence:** 3

**Summary:**

This paper studies whether self-supervised contrastive learning (CL) and a supervised counterpart, Negatives-Only Supervised Contrastive Learning (NSCL), produce similar representations during the training process. The authors theoretically prove that CL and NSCL maintain close alignment in their representation spaces under realistic conditions. They provide probabilistic bounds for alignment metrics such as centered kernel alignment (CKA) and representational similarity analysis (RSA), showing that representation similarity improves with larger numbers of classes, higher temperature, and certain batch size conditions. Empirical results confirm the theory: CL and NSCL representations become increasingly aligned as model scale and temperature grow, with NSCL tracking CL more closely than other supervised objectives.

**Strengths:**

1.	The theoretical proofs and analysis are thorough and well-founded, providing a clear explanation of the differences between contrastive learning and negative-only supervised learning.
2.	There is a strong alignment between the theoretical and empirical results.
3.	The paper is well-written, with a clear and accessible logic flow that is easy to follow

**Weaknesses:**

1.	My primary concern is the lack of clear practical insights derived from the theoretical analysis in this paper. For instance, [1] identifies that downstream classification performance depends on labeling errors and the connectivity of the augmentation graph, while [2] and its follow-up works analyze the role of negative samples in contrastive learning. However, in this paper, what practical benefits can we gain from the theoretical relationship between CL and NSCL, especially considering that we have no access to labeled data in self-supervised learning? In other words, what real-world applications or improvements in contrastive learning can be derived from the theoretical framework presented here?
2.	I'm uncertain whether the theoretical analysis offers additional advantages over existing work. Specifically, [2] also characterizes the training process of contrastive learning and establishes guarantees between contrastive loss (even when it is not optimal) and supervised downstream loss. The authors note that prior works often rely on restrictive assumptions, but several follow-up works have relaxed these assumptions, including the conditional independence assumption. It would be helpful to further discuss how the theoretical analysis in this paper improves to the existing body of work.
3.	Another weakness is the lack of connection between the terms in the theoretical bounds and the design choices in contrastive learning. For example, while the paper shows there is still a gap between CL and NSCL, it is unclear which specific terms in the bounds contribute to this gap. Additionally, how might we modify these terms to close the performance gap between the two methods?
4.	The empirical results show that the performance of CL and NSCL is still significantly below that of supervised contrastive learning, particularly on datasets like ImageNet. This raises the question of whether focusing only on supervised information in negative pairs limits the potential of the analysis. For instance, [3] directly compares the gap between contrastive learning and supervised contrastive learning. Besides, it is impractical to only use NSCL if we have access to supervised information.

[1] HaoChen J Z, Wei C, Gaidon A, et al. Provable guarantees for self-supervised deep learning with spectral contrastive loss[J]. Advances in neural information processing systems, 2021, 34: 5000-5011.

[2] Saunshi N, Plevrakis O, Arora S, et al. A theoretical analysis of contrastive unsupervised representation learning[C]//International conference on machine learning. PMLR, 2019: 5628-5637.

[3] Cui J, Huang W, Wang Y, et al. Rethinking weak supervision in helping contrastive learning[C]//International Conference on Machine Learning. PMLR, 2023: 6448-6467.

**Questions:**

See Weaknesses.

---

> ### Author Response · Authors · 2025-11-21
> **Addressing stated weaknesses**
>
> We thank the reviewer for the constructive and thoughtful comments. Below we respond to each one of the comments in the form of Q&A. Please see the full summary of changes above.
>
> > Reviewer: “My primary concern is the lack of clear practical insights derived from the theoretical analysis in this paper… In other words, what real-world applications or improvements in contrastive learning can be derived from the theoretical framework presented here?”
>
> **Answer**: Our goal is not to propose a new SSL algorithm or directly bound downstream accuracy, but to clarify when and how self-supervised CL behaves like a supervised method. Prior work (e.g., Luthra et al. 2025, Balestriero & LeCun 2024, Lee et al. 2025) linked CL and supervised objectives at the loss level; our contribution is to show that this connection also holds at the representation level, and to characterize which parameters control it (number of classes, batch size, temperature, learning rate schedule).
>
> We see this as providing a clearer mental model for how existing CL heuristics behave, rather than prescribing a specific new procedure. In particular, our results (and experiments) identify NSCL as the supervised objective that is most closely aligned with CL. Practically, this means that when a small labeled set or related labeled dataset is available, NSCL can be used as a faithful supervised proxy to study geometry, tune architectures/augmentations, and debug training dynamics for CL.
>
> – **Interpreting and tuning CL with a small amount of labels**: In realistic setups, one almost always has some labels (for a subset, or from a related labeled dataset). Our results show that NSCL is the supervised objective that stays most closely aligned with CL in representation space. This means NSCL can be used as a proxy tool to study geometry, pick augmentations/architectures, and debug training dynamics—then safely transfer those design choices to the fully self-supervised CL run.
>
> – **Explaining and justifying common CL heuristics**: The theory predicts—and our experiments confirm—that CL becomes more “supervised-like” as we increase dataset diversity (more classes), batch size, and temperature. This gives a principled explanation for why these design choices often help in practice and indicates regimes where CL should be expected to recover semantic, class-like structure even without labels.
>
> > Reviewer: “I'm uncertain whether the theoretical analysis offers additional advantages over existing work…”
>
> **Answer**: Many modern uses of representations—interpretability, segmentation, robustness, fairness—depend on the geometry of the embeddings (similarity structure, clustering), not just on downstream classification error. Guarantees of the kind in [2] (small contrastive loss implies small average supervised classification loss under a latent-class generative model) are very interesting for classification guarantees, but they do not by themselves ensure that CL models have good segmentation quality, interpretable features, or robustness; they only control classification risk.
>
> Our work takes concrete steps toward understanding geometric properties of CL representations (interpretability, segmentation, robustness, etc.) by studying how closely the actual CL training trajectory tracks a concrete supervised objective (NSCL) in representation space. This matters because, if the supervised counterpart has desirable geometric properties, our results indicate when CL will inherit them:
> 1. We give high-probability guarantees that CL and NSCL share essentially the same similarity structure (CKA/RSA) along the entire optimization path. Thus, whenever NSCL yields stable clusters, clean semantic neighborhoods, or more interpretable feature directions (and good classification), the coupled CL model is guaranteed to exhibit essentially the same representation geometry.
> 2. Our bounds make explicit how concrete knobs (number of classes, batch size, temperature, learning-rate schedule) control CL–NSCL alignment, explaining when CL should or should not behave like a supervised method and hence when this transfer of geometric properties is actually expected to hold.
> 3. We also identify NSCL—not generic CE or SCL—as the supervised objective most tightly coupled to CL. This gives practitioners with a small labeled set a principled proxy for tuning architectures/augmentations and auditing geometric properties, with our theory justifying that these design choices and diagnostics transfer to the unlabeled CL model.
> 4. We added a new experiment to show that the attention maps of CL and NSCL are more aligned with one another than the attention maps of CL and other supervised methods (Figs. 15-16). This shows that certain downstream tasks of the CL model like interpretability are transferred through NSCL. This is something that previous works that study classification guarantees are not able to capture.

---

> > ### Author Response · Authors · 2025-11-21
> >
> > > Reviewer: “Another weakness is the lack of connection between the terms in the theoretical bounds and the design choices in contrastive learning…”
> >
> > **Answer**: Our main similarity-space bound already decomposes the CL–NSCL gap into terms that correspond directly to standard CL design choices (number of classes, batch size, temperature, and learning-rate schedule). To make this more explicit, we added a paragraph after Thm. 1. We summarize the some of the dependencies here:
> >
> > 1. Larger number of classes $C$ → smaller $1/C$ smaller → smaller $\Delta_{\pi,\delta}(B;\tau)$ → smaller CL--NSCL gap.
> > 2. Larger batch size $B$ → smaller $\epsilon_{B,\delta}$ and $1/\sqrt{B}$, and the coefficient $\frac{1}{2\tau^2 B}$ in the exponential smaller →  smaller CL--NSCL gap.
> > 3. Larger temperature $\tau$ → factors $\frac{1}{\tau}$ and $\frac{1}{\tau^2}$ in the prefactor and exponential decrease → smaller CL--NSCL gap (as also observed empirically in Fig.4 and Fig.8).
> > 4. Smaller learning rates $\eta_t$ (or smaller total step size $\sum_t \eta_t$) → both the prefactor $\frac{1}{\tau\sqrt{B}}\sum_t \eta_t$ and the exponent $\exp\big(\frac{1}{2\tau^2 B}\sum_t \eta_t\big)$ decrease → smaller CL--NSCL gap.
> >
> > > Reviewer: “The empirical results show that the performance of CL and NSCL is still significantly below that of supervised contrastive learning, particularly on datasets like ImageNet…”
> >
> > **Answer**: We agree that NSCL is not the strongest supervised objective in terms of top-1 accuracy, and this is expected: on Mini-ImageNet, for example, SCL outperforms NSCL (≈76% vs. 72.6%). Our goal, however, is not to propose NSCL as the best supervised method, but to use it as a bridge between SSL and SL.
> >
> > Concretely:
> >
> > 1. NSCL is architecturally and structurally much closer to InfoNCE than SCL (it differs only in how same-class negatives are handled), and our theory shows that CL tracks NSCL very tightly in representation space. This makes NSCL a natural supervised “proxy” for understanding CL’s geometry, even if SCL achieves higher accuracy when labels are fully available.
> > 2. Focusing on supervised information in negative pairs therefore does not aim to close the CL–SCL performance gap directly; it aims to pick the supervised objective whose dynamics and similarity matrices are best coupled to CL, so that properties of NSCL (and its hyperparameters) meaningfully inform CL.
> > Regarding [3], their goal is different and complementary: they inject (weak) supervision into CL to move its performance closer to SCL. In contrast, we keep CL unchanged and study which supervised objective it implicitly mirrors (NSCL), and how standard design choices (batch size, temperature, learning rate, number of classes) control that alignment. Our work is about understanding and controlling the CL–SL relationship, not about replacing SCL (with NSCL) as the preferred supervised training objective when labels are available.

---

> > > ### Comment · Reviewer_nMEv · 2025-11-26
> > >
> > > Thanks for your detailed clarifications. The authors have addressed most of my concerns. However, I remain unsure about the practical applicability of the proposed theoretical analysis. Therefore, I have adjusted my score to 4.

---

> > > > ### Author Response · Authors · 2025-12-03
> > > > **Thank the reviewer for increasing their score and practical aspects**
> > > >
> > > > We thank the reviewer for increasing their score. To further explore the practicality of our work, we conducted a simple experiment in which we merge models directly in representation space. We found that combining the learned embeddings of a CL encoder with an NSCL encoder trained on only 30% of the labels already surpasses both the full-data CL model and the small-data NSCL model, reinforcing that NSCL and CL remain geometrically compatible in practice. We apply a simple interpolation technique; for full details, please refer to Appendix B.7 and Figure 17 in the Appendix of the revised version.
> > > >
> > > > This is practically useful in settings where a large self-supervised contrastive learning (CL) model is first pre-trained and later adapted for domain adaptation, fine-tuning, or semi-supervised pipelines. In such scenarios, NSCL provides a supervised variant that preserves the geometric properties of CL, allowing practitioners to incorporate supervision while still benefiting from large-scale self-supervised pre-training.

---

### Official Review · Reviewer_DFRi · 2025-10-30

**Soundness:** 4
**Presentation:** 3
**Contribution:** 3
**Rating:** 6
**Confidence:** 4

**Summary:**

The paper asks whether the objective-level affinity between contrastive learning (CL) and negatives-only supervised contrastive learning (NSCL) holds to the representation trajectories. Specifically, the authors bound the discrepancy between the cosine similarity matrices induced by CL and NSCL under shared randomness, and then derive bounds for CKA and RSA. Experiments with ResNet-50 support the predicted dependencies on the number of classes $C$, batch size $B$, and temperature $\tau$. In particular, CL aligns much more closely with NSCL than with SCL or cross-entropy.

**Strengths:**

1. A technically sound similarity-space analysis of representations. This is an incremental contribution over recent connections between CL and NSCL [2].
1. For large $B$ and $C$, the main high-probability result is simplified to
    $$
        \|\Sigma^{\mathrm{CL}}-\Sigma^{\mathrm{NSCL}}\|_F =  \tilde O\left(\frac{1}{\sqrt{B}}\left(\frac{1}{C} + \frac{1}{\sqrt{B}}\right)\right),
    $$
    ignoring the explicit dependence on $\tau$ and $T$. This in turn gives bounds on representation similarity for CL and NSCL. The empirical trends in $C$, $B$, and $\tau$ match the theory, and the CL-NSCL pair consistently shows stronger alignment than CL-SCL or CL-CE.

**Weaknesses:**

1. Appendix D explains why the proposed "similarity-descent" mirrors small-step gradient descent on parameters. Additional remark of that argument in the main text would help follow the logic from SGD updates to similarity dynamics.
1. The theory assumes class-balanced sampling. It would be useful to discuss how class imbalance alters the bound, for example by replacing $1/C$ with empirical class priors and indicating the resulting rate.
1. Prior work reports a non-trivial CKA between SimCLR and supervised ResNet-50 trained independently [1]. It would be great to have additional discussions on the extension of the current theoretical results to a broader class of supervised learning algorithms.
1. A comment on the tightness of the upper bound for $\|\Sigma^{\mathrm{CL}} - \Sigma^{\mathrm{NSCL}}\|_F$ would be helpful.

**Minor Comments**

1. Line 308: replace "GPUs" with "GPU".
1.  "ImageNet-1K" (line 84) vs "ImageNet-1k" (Figure 3 lower right)
1. The per-anchor CL and NSCL loss definitions would be improved for clarity.
1. Move the ``Additional notation for high-probability factors'' so it appears immediately before Theorem 1.

[1] Grigg, T. G., Busbridge, D., Ramapuram, J., \& Webb, R. (2021). Do self-supervised and supervised methods learn similar visual representations? arXiv:2110.00528.

[2] Luthra, A., Yang, T., \& Galanti, T. (2025). Self-Supervised Contrastive Learning is Approximately Supervised Contrastive Learning. arXiv:2506.04411.

**Questions:**

1. How does the analysis extend to class-imbalanced sampling?
1. Is the bound for $\|\Sigma^{\mathrm{CL}}-\Sigma^{\mathrm{NSCL}}\|_F$ tight?

---

> ### Author Response · Authors · 2025-11-21
> **Addressing stated weaknesses**
>
> We thank the reviewer for the constructive and thoughtful comments. Below we respond to each one of the comments in the form of Q&A. Please see the full summary of changes above.
>
> > Reviewer: “Appendix D explains why the proposed "similarity-descent" mirrors small-step gradient descent on parameters. Additional remarks of that argument in the main text would help follow the logic from SGD updates to similarity dynamics.”
>
> **Answer**: You are right that, taken in isolation, an exponential in the cumulative learning rate can in principle blow up. Our claim that “the difference stays small when the step sizes are not too aggressive” is meant in a scaled sense, and Eq. (9) makes this explicit. Without additional structural assumptions (e.g., stronger uniform curvature or smoothness conditions that further restrict the dynamics), one cannot generally hope for a qualitatively sharper dependence than the scaled exponential factor appearing in Eq. (9), so the behavior captured by the bound is essentially optimal at this level of generality.
> The exponential term is $\exp\bigl((1/(2\tau^2 B))\sum_{t=0}^{T-1}\eta_t\bigr)$, so the effective growth rate is scaled by $1/(2\tau^2 B)$ rather than by a bare constant. This implies:
> (i) If the cumulative step size is controlled relative to $\tau^2 B$, e.g., $\sum_t \eta_t \ll \tau^2 B$, then the exponent is close to $1$ and the exponential factor is benign. For typical contrastive-learning settings (large batches, $\tau \approx 0.5$–$1$, and a decaying learning rate), one has $\sum_t \eta_t = O(1\text{–}10)$ while $\tau^2 B = O(10^2\text{–}10^3)$, so the exponent is very small and the trajectories remain close.
> (ii) Conversely, if one uses extremely large or non-decaying learning rates so that $\sum_t \eta_t$ is comparable to or larger than $\tau^2 B$, the bound correctly becomes vacuous. This is precisely the regime we refer to as “overly aggressive,” where one should not expect the similarity and parameter trajectories to remain tightly coupled.
> Thus, the statement around line 1054 should be read as: the similarity and parameter trajectories stay close when the cumulative step size $\sum_t \eta_t$ is moderate relative to $\tau^2 B$. To clarify, we replaced the informal phrase “not too aggressive” with a more comprehensive discussion of the tradeoffs in the bound.
>
> > Reviewer: “The theory assumes class-balanced sampling. It would be useful to discuss how class imbalance alters the bound, for example by replacing 1/C with empirical class priors and indicating the resulting rate.”
>
> **Answer**: Following the reviews, we updated the paper to include imbalanced classes. To make our experiments more extensive, we added a new experiment with SVHN, a well-known class-imbalanced classification task (Fig. 14).
>
> > Reviewer: “Prior work reports a non-trivial CKA between SimCLR and supervised ResNet-50 trained independently [1]...”
>
> **Answer**: Thank you for pointing us to this work. We have added a citation to it in the revised manuscript (lines 124-126). In fact, we see it as further strengthening the motivation for our paper. As discussed in lines 124-128, there is growing evidence that SSL and SL can learn closely related representations ([1], Luthra et al. 2025, Balestriero & LeCun 2024, Lee et al. 2025), but it remains unclear which supervised objectives are most similar to contrastive SSL and in what sense. The paper you mention provides empirical evidence for alignment between SimCLR and a CE-trained ResNet-50 via CKA. In contrast, our theory shows that NSCL is provably aligned with CL, and our experiments systematically compare NSCL to several SL methods, demonstrating that NSCL is consistently more aligned with CL than other SL baselines and that in some cases CL fails to align with CE minimization. Since CL is aligned with CE only in special circumstances, we believe that extending our work to a broader class of supervised learning algorithms would be valuable and interesting but beyond the current framework.
>
> > Reviewer: Typos.
>
> **Answer**: We updated the manuscript to fix these issues.
>
> > Reviewer: “How does the analysis extend to class-imbalanced sampling?”
>
> **Answer**: We address this question in “Weaknesses” above.

---

> > ### Author Response · Authors · 2025-11-21
> >
> > > Reviewer: “A comment on the tightness of the upper bound for $|\sum^{\text{CL}} - \sum^{\text{NSCL}}|_F$ would be helpful.”
> >
> > > “Is the bound for $|\sum^{\text{CL}} - \sum^{\text{NSCL}}|_F$ tight?”
> >
> > **Answer**: We do not expect this bound to be tight. As is common in machine learning theory (e.g., Bousquet & Elisseeff, 2002; Hardt et al., 2016; Mou et al., 2018; Kuzborskij & Lampert, 2017), our worst-case guarantee is conservative and can be far from equality in realistic settings. Its purpose is to capture which parameters matter and how the Frobenius norm of the similarity difference scales with them, rather than to provide a numerically sharp estimate. We now clarify in the text that the result should be interpreted as a conceptual worst-case guarantee rather than a tight bound.

---

> > > ### Comment · Reviewer_DFRi · 2025-11-26
> > >
> > > I thank the authors for the detailed clarifications and updated results. My concern about class imbalance is now resolved, as the authors generalized the theory and added supporting experiments.
> > >
> > > I want to clarify one minor point. In Weakness 1, my concern was not about the exponential factor blowing up, but about the "similarity-descent" surrogate whose justification is deferred to Appendix D. For readability, I still recommend adding a few sentences in the main text (for example around line~219) summarizing "parameter descent acts like similarity descent" from Appendix D.

---

> > > > ### Author Response · Authors · 2025-11-29
> > > > **Thank you again for your helpful clarification and for your continued appreciation of our work**
> > > >
> > > > Thank you again for your helpful clarification and for your continued appreciation of our work. In response to your comment, we have uploaded a revised version of the paper in which we added a brief summary in the main text (lines 218–228) of the justification provided in Appendix D, explaining why parameter descent acts like similarity descent.

---

### Official Review · Reviewer_Dkwy · 2025-11-01

**Soundness:** 2
**Presentation:** 3
**Contribution:** 3
**Rating:** 6
**Confidence:** 2

**Summary:**

This work provides an in-depth theoretical analysis of the alignment between Self-supervised Contrastive Learning (CL) and Negative-only Supervised Contrastive Learning (NSCL). In particular, a bound is derived coupling the CL and NSCL similarity dynamics, demonstrating the CL-NSCL behavioral alignment. Several experiments are also conducted to empirically validate the findings.

**Strengths:**

Leveraging similarity-space dynamics, this paper bridges CL and NSCL behavior in a theoretically rigorous manner. Empirical verifications are also done thoroughly, with near real-world scale datasets like tiny-ImageNet. Empirical results support theoretical findings.

**Weaknesses:**

As the authors stated in the conclusion/limitation section, the proposed bounds can be loose under large-scale settings due to the exponential factors on cumulative step size.

More importantly, please see my question below regarding the discrepancy between similarity-based dynamics and parameter-based dynamics, which I think is a very important step in the derivation of the proposed bound. It perhaps deserves a more detailed explanation.

**Questions:**

In line 1054 (Appendix D), the authors stated that "the difference between the two trajectories stays small when the step sizes are not too aggressive." However, Eq. (9) contains an exponential term on cumulative learning rate—does this really guarantee that the trajectories stay close even if the step sizes are not aggressive? Even so, how is this effect accounted for in Thm. 1, while practical algorithms perform parameter-GD instead of similarity-GD?

Intuitively, the similarity-GD and parameter-GD differ significantly as shown in Eq. (5) ($P_t := J_t J_t^\top$). The update delta of each similarity entry, when trained via parameter-GD, can vary depending on the network gradient evaluated at different input points. I find it somewhat counter-intuitive that the results on similarity-GD can be applied to algorithms utilizing parameter-GD. Therefore, It would be glad to hear the authors' responses on this matter (or please correct me if I misunderstood anything).

---

> ### Author Response · Authors · 2025-11-21
> **Addressing stated weaknesses**
>
> We thank the reviewer for the constructive and thoughtful comments. Below we respond to each one of the comments in the form of Q&A. Please see the full summary of changes above.
>
> > Reviewer: As the authors stated in the conclusion/limitation section, the proposed bounds can be loose under large-scale settings due to the exponential factors on cumulative step size.
>
> **Answer**: Indeed, we do not expect this bound to be tight. As is very common in machine learning theory, our worst-case guarantee is conservative and can be far from equality on realistic data. Its purpose is to capture which parameters matter and how the similarity-matrix gap scales with them, rather than to provide a numerically sharp estimate. We note that many influential results in optimization and stability theory for deep learning also rely on loose worst-case bounds (see Bousquet & Elisseeff, 2002; Hardt et al., 2016; Mou et al., 2018; Kuzborskij & Lampert, 2017), yet still provide useful conceptual guidance.
> Furthermore, without additional structural assumptions (e.g., stronger uniform curvature or smoothness conditions that further restrict the dynamics), one cannot generally hope for a qualitatively sharper dependence than the scaled exponential factor appearing in Eq. (9), so the behavior captured by the bound is essentially optimal at this level of generality.
>
> > Reviewer: In line 1054 (Appendix D), the authors stated that "the difference between the two trajectories stays small when the step sizes are not too aggressive." However, Eq. (9) contains an exponential term on cumulative learning rate—does this really guarantee that the trajectories stay close even if the step sizes are not aggressive?
>
> Answer: You are right that, taken in isolation, an exponential in the cumulative learning rate can in principle blow up. Our claim that “the difference stays small when the step sizes are not too aggressive” is meant in a scaled sense, and Eq.~(9) makes this explicit. The bound there has the form
> $$
> \|\widehat\Sigma_T - \widetilde\Sigma_T\|_F \le \exp\Bigl(\frac{1}{2\tau^2 B}\sum_t^{T-1}\eta_t\Bigr) \Gamma_T,
> $$
> where $\Gamma_T$ is a term that is linear in $\sum_t \eta_t$ plus a term that is quadratic in $\sum_t \eta_t^2$. Thus, the effective growth rate in the exponential is scaled by $1/(2\tau^2 B)$ rather than by a bare constant. Without additional structural assumptions (for example, stronger curvature or smoothness conditions that further restrict the dynamics), one cannot generally hope for a qualitatively sharper dependence than this scaled exponential factor, so the behavior captured by the bound is essentially optimal at this level of generality.
>
> The exponential term is $\exp\bigl((1/(2\tau^2 B))\sum_{t=0}^{T-1}\eta_t\bigr)$, so the effective growth rate is governed by the normalized cumulative step size $\sum_t \eta_t / (\tau^2 B)$. This implies: (i) if the cumulative step size is controlled relative to $\tau^2 B$, e.g., $\sum_t \eta_t \ll \tau^2 B$, then $(1/(2\tau^2 B))\sum_t \eta_t$ is small and the exponential factor is close to $1$, and hence benign. This is precisely the regime we make explicit in Appendix~D, where we assume that $\sum_t \eta_t / (\tau^2 B)$ and $\sum_t \eta_t^2$ are both bounded. In typical contrastive-learning settings (large batches, $\tau \approx 0.5$--$1$, and a decaying learning rate), one has $\sum_t \eta_t$ on the order of $1$ to $10$, while $\tau^2 B$ is on the order of $10^2$ to $10^3$, so the exponent is very small and the similarity and parameter trajectories remain close. (ii) Conversely, if one uses extremely large or non-decaying learning rates so that $\sum_t \eta_t$ is comparable to or larger than $\tau^2 B$, the bound correctly becomes vacuous. This is precisely the regime we refer to as “overly aggressive,” where one should not expect the trajectories to remain tightly coupled.
>
> Thus, the statement around line 1054 should be read as: the similarity and parameter trajectories stay close when the normalized cumulative step size $\sum_t \eta_t / (\tau^2 B)$ is moderate and the schedule is such that $\sum_t \eta_t^2$ remains bounded. To clarify this point, we replaced the informal phrase “not too aggressive” with a more detailed discussion of these tradeoffs in the bound.
>
> To address the reviewer’s concern, we evaluate our bound values for ImageNet-1k where we use the following practical configuration (C=1000, B=1024, $\tau$=0.1). Since our bound also depends on the total number of epochs (T), we report the bound for different values of T as well as learning rates.
>
> | $\eta$ | **T=100** | **T=300** | **T=500** | **T=1000** |
> | :--- | :--- | :--- | :--- | :--- |
> | $1\times10^{-3}$ | 15.36 | 46.01 | 76.97 | 155.67 |
> | $1\times10^{-4}$ | 1.53 | 4.57 | 7.61 | 15.22 |

---

> > ### Author Response · Authors · 2025-11-21
> >
> > > Reviewer: Even so, how is this effect accounted for in Thm. 1, while practical algorithms perform parameter-GD instead of similarity-GD?
> >
> > **Answer**: In order to translate Thm. 1 into a bound for the parameter-GD, we can apply the analysis in the appendix twice and get a bound on the gap between the CL and NSCL similarity matrices that is the same as the one in Thm. 1 + two times the bound in the Eq. 11. We added a comment on this in lines 251-255.
> >
> > > Reviewer: Intuitively, the similarity-GD and parameter-GD differ significantly as shown in Eq. (5). The update delta of each similarity entry, when trained via parameter-GD, can vary depending on the network gradient evaluated at different input points. I find it somewhat counter-intuitive that the results on similarity-GD can be applied to algorithms utilizing parameter-GD. Therefore, It would be glad to hear the authors' responses on this matter (or please correct me if I misunderstood anything).
> >
> > **Answer**: We agree that parameter–GD and similarity–GD need not coincide in general, and Eq. (5) is precisely where this discrepancy appears: parameter–GD induces a preconditioned update in similarity space plus a higher-order remainder. This is a worst-case bound; in practice the discrepancy is typically even smaller than what the theory guarantees. Our analysis does not ignore this gap; Eq. (5), together with the Jacobian/smoothness assumptions, is exactly what we use to control it.
> > More concretely, Eq. (5) decomposes the similarity update into (i) a preconditioned gradient step, and (ii) a Taylor remainder that is quadratic in the step size. The assumptions on the Jacobian ensure that $P_t$ has uniformly bounded operator norm and that the remainder stays small. Unrolling this over time gives an explicit bound on the distance between the similarity trajectory induced by parameter–GD and the “ideal” similarity–GD trajectory; this bound depends only on the normalized cumulative step size and the sum of squared step sizes, with a stabilizing $1/\sqrt{B}$​ and temperature dependence.
> > In the same regime where practitioners already keep training stable—moderate learning rates, decaying schedules, large batch sizes, and non-extreme temperatures—the normalized cumulative step size and $\sum_t \eta^2_t$​ remain bounded, so the bound guarantees that the two similarity trajectories stay close. This is why Theorem 1, though stated for similarity–GD, is still informative for practical algorithms using parameter–GD: the effect highlighted in Eq. (5) is explicitly accounted for, and the resulting discrepancy is provably small under standard hyperparameter choices.

---

### Official Review · Reviewer_SXMf · 2025-11-01

**Soundness:** 3
**Presentation:** 3
**Contribution:** 3
**Rating:** 8
**Confidence:** 3

**Summary:**

While contrastive learning and negatives-only supervised contrastive learning have similar objectives, this does not necessarily imply that they will learn similar representations through training. Using CKA and RSA, the authors analyze this alignment's emergence in various conditions to provide a better understanding of the similarities between such methods.
This is complemented by clear theoretical results that provide intuition on each parameter's influence on this alignment.

**Strengths:**

- Beyond training objective similarity, the authors provide an analysis of representation similarity through training, providing better insights in similarities between CL and NSCL.

- On top of the thorough experiments, the authors prove clear theoretical results that help guide empirical analysis.

- The detailed explanation of the bound obtained in Theorem 1 lines 246-262 is greatly appreciated to provide more practical intuitions in the result.

- The overall work strengthens our understanding of similarities and differences between CL and related supervised learning algorithms.

- Experiments are done at a good practical scale (up to ImageNet using a resnet 50) which helps yield more practically relevant insights.

**Weaknesses:**

1) My main issue is that it is unclear whether alignment is beneficial in practice. Having CL be more aligned with a **fixed** high performing NSCL model is most likely good in practice. However in the scenarios where alignment increases, it is possible that performance suffers as well. It would be interesting to have more data points regarding alignment vs performance potential tradeoffs.

2) While alignment is studied in the general case, it seems that it can get worse in practical settings.
Line 415-146 “Alignment increases with higher values of $\tau$”. Usually for CL temperatures used tend to be low, with performance decreasing as it is increases, suggesting that increasing alignment through temperature comes at a practical cost.

3) Line 079: $C$ is discussed but not introduced yet (I assume it is the number of classes from line 125). It would be good to have its meaning be clear when first discussed.

**Questions:**

1) In Figure 2 there seems to be a dynamics change around 10-100 epochs when looking at SCL and CE, any intuition as to why ?

---

> ### Author Response · Authors · 2025-11-21
> **Addressing stated weaknesses**
>
> We thank the reviewer for the positive feedback and their thoughtful recognition of our theoretical analysis, empirical scale, and the practical intuitions provided.
>
> > Reviewer: “My main issue is that it is unclear whether alignment is beneficial in practice. Having CL be more aligned with a fixed high performing NSCL model is most likely good in practice. However in the scenarios where alignment increases, it is possible that performance suffers as well. It would be interesting to have more data points regarding alignment vs performance potential tradeoffs.”
>
> Answer: We agree that alignment alone does not guarantee good performance. Our results are best interpreted as follows: for a given architecture and dataset, a configuration is promising for SSL if (1) the same optimization setup works well in supervised mode (i.e., with the same optimizer, LR schedule, batch size, temperature, etc., NSCL attains good performance), and (2) under that setup, CL is well aligned with NSCL in representation space.
> Recent work (e.g., Luthra et al. 2025, Balestriero & LeCun 2024, Lee et al. 2025) addresses (2) at the loss level, arguing that CL carries a supervised-like signal. Our contribution is to move from the loss to the representations and to quantify how standard “knobs” (batch size, temperature, learning-rate schedule, number of classes) control when this alignment actually holds in practice.
> To directly address the reviewer’s request, we added experiments (Figs. 12-13) that relate alignment and performance more explicitly. These show that:
> 1. when a given configuration trains NSCL to high accuracy, increasing CL–NSCL alignment shrinks the performance gap between CL and NSCL;
> 2. when the same configuration does not train NSCL well (e.g., overly aggressive hyperparameters), CL still becomes well aligned to that weaker NSCL solution, and both models perform poorly.
> Thus, alignment is beneficial conditional on the supervised run being good. Our results are best viewed as: given an optimization/configuration that already yields a strong supervised model, our bounds and experiments indicate when CL will track that supervised solution closely (and can be expected to perform well), and when misaligned configurations are likely to hurt SSL.
>
> > Reviewer: “While alignment is studied in the general case, it seems that it can get worse in practical settings. Line 415-146 “Alignment increases with higher values of tau”. Usually for CL temperatures used tend to be low, with performance decreasing as it increases, suggesting that increasing alignment through temperature comes at a practical cost.”
>
> Answer: Thank you for this observation. Our statement about $\tau$ is descriptive, not prescriptive: higher $\tau$ makes the CL and NSCL representations more aligned, which our theory predicts and our experiments confirm.
>
> We agree that higher temperatures can hurt the absolute performance of CL/NSCL models. To address this concern directly, Figure 6 (added to the appendix) shows that while alignment increases with $\tau$ and the CL–NSCL accuracy gap narrows, the absolute performance of both models can decrease at high $\tau$. Our goal is to explain the mechanism driving alignment, not to recommend using very large temperatures in practice.
>
> > Reviewer: “Line 079:  C is discussed but not introduced yet (I assume it is the number of classes from line 125). It would be good to have its meaning be clear when first discussed.”
>
> Answer: We thank the reviewer for pointing this out. The variable $C$ refers to the number of hidden classes. We have updated the paper to define $C$ at its first appearance.

---

> > ### Author Response · Authors · 2025-11-21
> > **Figure 2 intuition**
> >
> > > Reviewer: “In Figure 2 there seems to be a dynamics change around 10-100 epochs when looking at SCL and CE, any intuition as to why ?”
> >
> > **Answer**: We agree that there is a noticeable change in dynamics around 10–100 epochs for SCL and CE. Our intuition is that all three objectives—SCL, CE, and NSCL—encourage features to cluster by class (neural collapse), but they differ in how strongly and how early they push toward that regime. SCL provides the most direct incentive: it explicitly pulls together samples (and augmentations) from the same class while pushing apart samples from different classes, so it tends to induce neural collapse earlier in training. CE does this more indirectly, by enforcing class margins and, through regularization, shrinking feature norms, which typically leads to a slower progression toward collapse.
> > In the self-supervised setting, we do not expect the SSL representations to become as tightly collapsed as the supervised ones, since they are learned without labels. As training progresses into the 10–100 epoch range, the supervised SCL and CE models begin to move more aggressively into the neural-collapse regime, while the SSL model continues to optimize a label-free objective. This mismatch in how quickly and how strongly the supervised and self-supervised models collapse their features naturally leads to a change in the alignment dynamics over that interval, which is exactly what we observe in Figure 2.
> > Following the reviews, we added a discussion in lines 413–430 to provide intuition for why this behavior occurs in our experiments.

---

> > > ### Comment · Reviewer_SXMf · 2025-11-26
> > >
> > > Thank you for the answers and clarifications, as the added results and discussions help make clearer the links to practice. As my score was already positive, I will leave it as is.

---

### Author Response · Authors · 2025-11-21
**Summary of Changes**

### Summary of Changes

1. Added Fig. 12 and Fig. 13 that directly relates CL–NSCL alignment to performance, showing:

(i) When a configuration trains NSCL to high accuracy, increased CL–NSCL alignment shrinks the performance gap between CL and NSCL.

(ii) When the same configuration yields a weak NSCL model (e.g., overly aggressive hyperparameters), CL still aligns to this weaker solution and both models perform poorly.

2. Added Fig. 12 and Tab. 2 studying the effect of temperature $\tau$, showing that:

(i) Alignment increases and the CL–NSCL accuracy gap narrows as $\tau$ increases.

(ii) The absolute accuracy of both CL and NSCL can drop at very high temperatures.

3. We replicated the experiments in Figs. 2-5 for ViT models in Figs. 7, 8, 9, 11.

4. Added new experiments in Figs. 15-16 showing that the attention maps of CL are more similar to the attention maps of NSCL compared to SCL and CE. See details in Sec. B.5.

6. Clarified that our Frobenius-norm bound on the similarity gap is a conservative worst-case guarantee, whose purpose is to capture how key parameters (batch size, temperature, number of classes, learning-rate schedule) influence the CL–NSCL similarity gap, rather than to provide a numerically tight estimate (see lines 520-530).

7. Expanded the discussion on why similarity-descent mirrors small-step parameter gradient descent, and clarified the conditions under which the trajectories remain tightly coupled (lines 1395-1411).

8. Evaluated the bound numerically on ImageNet-1k for realistic hyperparameters (e.g., $C=1000$, $B=1024$, $\tau=0.1$) and different numbers of epochs and learning rates, illustrating the magnitude of the bound in practical regimes.

9. Clarified how Thm. 1, stated for similarity-GD, extends to parameter-GD (lines 251-255).

10.  Added an explanation of the change in dynamics around 10–100 epochs in Fig. 2 (lines 413-430).

11. Extended the theoretical analysis to class-imbalanced data, replacing uniform class factors with empirical class priors and explaining the resulting changes in the rate.

12. Added new experiments on SVHN, a naturally class-imbalanced dataset, to empirically validate the theory in the imbalanced setting (Fig. 14).

13. We expanded the contributions section to clarify: (i) that our theory does not propose a new algorithm or bound downstream accuracy, but instead characterizes when self-supervised CL behaves like the supervised NSCL objective in representation space. (ii) Added discussion explaining how our representation-level guarantees complement prior work that links CL and supervised objectives at the loss level, especially for applications that depend on representation geometry (interpretability, etc.) rather than only downstream classification error (lines 84-93).

14. Made explicit how each term in our main bound corresponds to standard CL “knobs” and how changing them strengthens the alignment between CL and NSCL (lines 235-250).

15. Added a citation to work showing non-trivial CKA alignment between SimCLR and supervised ResNet-50 (lines 124-126). To strengthen our motivation, we added a citation to Lee et al. 2025 (lines 126-128).

16. Improved the per-anchor CL and NSCL loss definitions for clarity (lines 152-171). And fixed some typos mentioned by the reviewers.

17. Moved the “Additional notation for high-probability factors” immediately before Thm. 1 (lines 222-227).

---

### Meta-Review · Area_Chair_hSLQ · 2026-01-02

**Summary:**

This paper studies the connection between self-supervised contrastive learning (CL) and negatives-only supervised contrastive learning (NSCL). Although CL and NSCL share similar loss formulations, the authors go beyond this observation and investigate whether these two learning paradigms lead to similar representations during training. First, the authors empirically analyze the alignment between CL and NSCL using CKA and RSA, providing a deeper understanding of the representational similarity between the two methods. They then derive a theoretical bound that explains this alignment behavior and offers further intuition regarding how various parameters and factors influence the degree of alignment.

**Reviewer Concerns:**

All reviewers agree that this is a strong paper with interesting empirical observations and solid theoretical contributions. The theoretical insights are also valuable. Several questions were raised during the review process. Reviewer SXMf asked about the potential negative impact of such alignment. Reviewer Dkwy raised an insightful question regarding the exponential term involving the cumulative learning rate when bounding the difference between two optimization trajectories. The authors provided a comprehensive response. While this exponential term may indeed make the bound loose in some cases, the authors presented concrete examples to justify its necessity, and I do not view this as a serious concern. Reviewer DFRi asked for clarifications and requested additional experiments on imbalanced class settings, which were properly addressed by the authors. Reviewer nMEv initially gave a low score of 2 but increased it to 4 after the rebuttal; their main concern was the practical implications of the theoretical results. I do not consider this a major issue, as the theory itself offers an interesting and meaningful perspective on the alignment between CL and NSCL.

Overall, after carefully reading the paper, I agree with the reviewers that this is a solid theory paper with meaningful contributions to contrastive learning. Therefore, I vote to accept the paper and recommend that the authors incorporate the reviewers’ suggestions in the final version.

**Reviewer Scores:**

Three reviewers are positive about the paper, with scores of 8, 6, and 6. The remaining reviewer increased their score from 2 to 4 but still expressed concerns regarding practical applicability and is unlikely to further increase the score. However, I do not consider this to be a critical concern.

---

### Decision · Program_Chairs · 2026-01-26

Accept (Poster)